# A highly proton conductive perfluorinated covalent triazine framework via low-temperature synthesis

Lijiang Guan[1], Zhaoqi Guo[2], Qi Zhou[1], Jin Zhang[1], Cheng Cheng[1], Shengyao Wang [3], Xiang Zhu[4], Sheng Dai [5] & Shangbin Jin [1] ✉

Proton-conducting materials are essential to the emerging hydrogen economy. Covalent triazine frameworks (CTFs) are promising proton-conducting materials at high temperatures but need more effective sites to strengthen interaction for proton carriers. However, their construction and design in a concise condition are still challenges. Herein, we show a low temperature approach to synthesize CTFs via a direct cyclotrimerization of aromatic aldehyde using ammonium iodide as facile nitrogen source. Among the CTFs, the perfluorinated CTF (CTF-TF) was successfully synthesized with much lower temperature ( $\leq 160\,°C$ ) and open-air atmosphere. Due to the additional hydrogen-bonding interaction between fluorine atoms and proton carriers ($H_3PO_4$), the CTF-TF achieves a proton conductivity of $1.82 \times 10^{-1}\,S\,cm^{-1}$ at 150 °C after $H_3PO_4$ loading. Moreover, the CTF-TF can be readily integrated into mixed matrix membranes, displaying high proton conduction abilities and good mechanical strength. This work provides an alternative strategy for rational design of proton conducting media.

Currently, the proton-conducting materials working above 100 °C show many great advantages in proton-exchange membrane fuel cells[1–3]. Proton conducting materials need to have high durability and strong interaction with proton carriers (i.e. $H_3PO_4$)[4,5]. In recent years, as an emerging class of porous organic polymers with high chemical stability, porosity and the alkaline pyridinic nitrogen content, covalent triazine frameworks (CTFs) have been explored as promising proton-conducting materials[6–9]. The regulation of CTFs in proton conduction at high temperatures is mainly focused on heterocyclic nitrogen and aromatic structures. To enhance the proton conductivity, it is important to introduce as more hydrogen acceptors as possible, which may tune the interaction sites in the CTFs with proton carriers. However, due to the limitation of synthesis methods and the building blocks, the introduction of

other precise hydrogen acceptors for proton conduction has not been widely explored.

The fluorine atom can form hydrogen bond with hydrogen atoms and has many unique effects in functional properties due to its strong electronegativity and robust bonding effect[10,11]. The introduction of fluorine atoms in polymers can bring many benefits for applications[12], i.e. increase the stability and facilitate the proton migration in the proton conducting polymers (i.e. Nafion)[4,13]. While in CTFs, the introduction of fluorine atoms may also benefit for high temperature proton conduction because it may provide more hydrogen-bonding interaction sites between the CTF and the proton carriers. In particular, perfluorinated CTFs with high fluorine content could exhibit the greatest number of anchoring sites with proton carriers in the framework structures. However, despite some fluorinated CTFs have been

[1]School of Chemical Engineering and Technology, Xi'an Jiaotong University, No. 28, Xianning West Road, Xi'an, Shaanxi 710049, China. [2]School of Chemical Engineering, Northwest University, No.229 Taibai North Road, Xi'an, Shaanxi 710069, China. [3]College of Science, Huazhong Agricultural University, Wuhan 430070, China. [4]State Key Laboratory for Oxo Synthesis and Selective Oxidation, Suzhou Research Institute of Lanzhou Institute of Chemical Physics, Chinese Academy of Sciences, Lanzhou 730000, China. [5]Chemical Sciences Division, Oak Ridge National Laboratory, Oak Ridge, TN 37831, USA. ✉ e-mail: shangbin@xjtu.edu.cn

synthesized, which demonstrate the merits for $CO_2$ uptake and energy storage[14–18], the synthesis of perfluorinated CTFs with both high fluorine content and surface area have been rarely reported. Previously, the fluorinated CTFs were obtained through the ionothermal strategy in the presence of $ZnCl_2$ at high temperature (400 °C), which lead to cleavage of the C−F bonds and low fluorine content[14,17,18]. More recently, the perfluorinated CTF was able to be prepared via ionothermal condition from a nitrile monomer using crafted catalyst under a relatively high temperature (275 °C)[19]. The synthesis of perfluorinated CTFs was also reported by superacid-catalysis at high temperature (250 ~ 350 °C)[20]. These methods have advanced the synthesis and application of fluorinated CTFs. However, the syntheses still require very high temperature in the sealed systems. The high temperature synthesis is not favorable to achieve high fluorine content or retention of the triazine structures. Construction of perfluorinated CTFs with high fluorine content (F content > 30 wt%) under more gentle conditions remains a challenge.

Herein, we successfully established an approach to synthesize CTFs via direct cyclotrimerization of aromatic aldehydes using $NH_4I$ as facile nitrogen source (Fig. 1). The conditions of this method are much milder, using low temperature (160 °C), open-air atmosphere, readily available raw materials, and relatively low catalyst dosage. We have found that the gentle approach is effective to synthesize CTFs with various structures. Particularly, the synthetic approach enables the synthesis of perfluorinated CTF with high fluorine content and surface area under much milder conditions as compared to the

other reported methods (Supplementary Fig. 1 and Supplementary Table 1)[14,17–20].

The resulting CTFs display promising proton conduction properties after binding with the $H_3PO_4$ with the merits of rich nitrogen content and high stability[6–9]. In particular, the electronegative fluorine sites together with the triazine units in the perfluorinated CTF (CTF-TF), which provide the precise interaction sites, can effectively lock $H_3PO_4$ and act as hydrogen bond acceptor to facilitate proton transport. Consequently, the CTF-TF loading with $H_3PO_4$ delivers a proton conductivity of $1.82 \times 10^{-1}\,S\,cm^{-1}$ at 150 °C, which ranks the highest value among the reported CTFs and is also comparable to other porous organic polymer conductors. Moreover, owing to the lamellar structures, the CTFs can be further processed into mixed matrix membranes (MMMs) by filling into commercial poly(vinylidene fluoride) (PVDF) polymer. The resultant membranes not only display high proton conductivities after doping with $H_3PO_4$, but also exhibit good mechanical strengths. This work presents a good paradigm to endow fluorinated CTF with high proton conduction and provides a promising way to extend the functionality and applications of the CTFs.

## Results
### Synthesis and characterization of CTFs
To test the feasibility of cyclotrimerization of aromatic aldehydes to prepare CTFs with $NH_4I$ as nitrogen source, the synthesis process of CTF-1 was optimized in different conditions. Effects of different

Fig. 1 | **Preparation of CTFs. a** Proposed reaction mechanism for synthesis of triazine unit; **b** Schematic synthesis of CTFs.

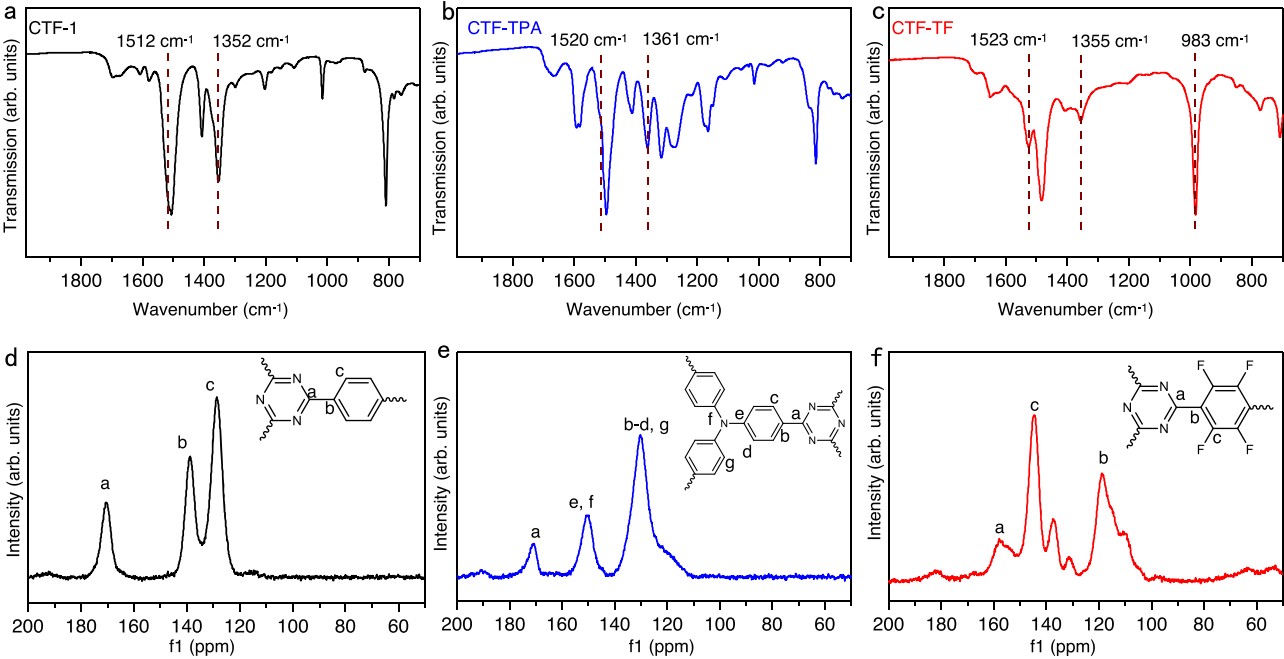

**Fig. 2 | Characterization of CTFs. a–c** FT-IR spectra and **d–f** solid-state[13]C CP-MAS NMR spectra of CTF-1 (black), CTF-TPA (blue), and CTF-TF (red).

reaction conditions, i.e., the catalyst, solvent, and reaction temperature, were investigated (Supplementary Table 2). We found that controlling the starting reaction temperature is crucial for the synthesis of CTF-1 because the decomposition of NH₄I easily occurred at higher temperatures and the catalyst promotes the formation of triazine units by redox reaction with iodide ions. The model reaction could also be successfully carried out under the same optimized condition (Supplementary Fig. 2), which further indicates the feasibility of the method[21]. The other two CTFs, named as CTF-TPA and CTF-TF, were successfully generated from aldehyde monomers under optimal conditions, which demonstrates the universality of this new method (Fig. 1). These results show that this facile strategy could be used to efficiently synthesize diversified functionalized CTFs, particularly for the perfluorinated CTF under much lower temperature condition.

Fourier-transformed infrared (FT-IR) spectroscopy, solid-state cross-polarization magic angle spinning carbon-13 nuclear magnetic resonance (CP-MAS [13]C-NMR), and X-ray photoelectron spectroscopy (XPS) were used to confirm the formation of CTFs. As shown in Fig. 2a–c and Supplementary Fig. 3, the characteristic vibration signals of triazine units are clearly observed. Typically, in the CTF-1 the two sharp peaks derived from triazine units are at 1352 cm⁻¹ and 1512 cm⁻¹, while the two peaks of CTF-TPA shift to 1361 cm⁻¹ and 1520 cm⁻¹, respectively[22–27]. Two peaks at 1355 cm⁻¹ and 1527 cm⁻¹ were also found in CTF-TF, which are corresponding to the vibrations of triazine rings. Furthermore, the signal at 983 cm⁻¹ could be assigned to stretching bands of C–F, which reveals the formation of CTF-TF[28]. In the CP-MAS [13]C-NMR spectrum, the CTF-1 demonstrates the obvious carbon signals from the triazine units at 170.4 ppm and the phenyl carbons at 128.7 ppm and 138.8 ppm (Fig. 2d)[24,25,29]. The spectrum of CTF-TPA also displays a peak at 170.7 ppm, assigned to the carbon in the triazine units (Fig. 2e). Furthermore, the peak at 150.8 ppm corresponds to the carbon atoms that are directly connected to the nitrogen. Meanwhile, the broad peak at 130.3 ppm is corresponding to the aromatic ring in the CTF-TPA[30]. In the CTF-TF, the carbon signal of triazine rings shifts to 159 ppm because of the introduction of fluorine atoms[31,32], and the carbon signal of fluorinated benzene rings are located at 130–145 ppm (Fig. 2f)[19,32]. Moreover, the carbon atoms connected with triazine rings display a peak at 119 ppm[19]. According to XPS measurements (Supplementary

Fig. 4), the major peak at 399.2 eV in CTF-1 and CTF-TF was attributed to the N species in the triazine units[33]. In the CTF-TPA, the peak at 398.7 eV belongs to nitrogen of the triazine ring, whereas the signal at 400.1 eV corresponds to the nitrogen atoms from the triphenylamine moiety[30]. The elemental analysis (EA) shows that the carbon and nitrogen contents of resulting CTFs present with reasonable contents, and F content of CTF-TF is as high as 30.2 wt% (Supplementary Table 3), which are smaller than ideal values calculated from crystalline structures due to the amorphous network structures[26,33,34]. The fluorine content of CTF-TF is comparable to the highest values reported so far and is much higher than that of most fluorinated CTFs (Supplementary Table 1). Notably, this marks an instance of synthesizing a type of perfluorinated CTF with a significantly high fluorine content through such gentle methods. Previous conventional approaches struggled to achieve this level of perfluorination under mild conditions.

Powder X-ray diffraction (PXRD) measurements (Supplementary Fig. 5) indicated that the CTF-1 displays a low broad peak at 7.5°, suggesting it only contains low crystallinity to some degree[33,34], whereas CTF-TPA and CTF-TF are amorphous. The porosities of CTFs were assessed by N₂ adsorption-desorption measurements at 77 K (Supplementary Fig. 6). CTF-1 and CTF-TF exhibit obvious adsorption of nitrogen in the low-pressure region of P/P₀ < 0.1, which suggests that the two materials have a large number of microporous structures (Supplementary Fig. 6a, c). By Non-local density functional theory (NL-DFT) calculation, the pore size distributions show that the dominant pore sizes for CTF-1 is ~1.2 nm and that of CTF-TF is about 1.1 nm (Supplementary Fig. 6d, f). Additionally, owing to particle accumulation and defects, unexpected mesoporous and macropore also exist in the CTFs. The adsorption of nitrogen by CTF-TPA in the low-pressure region decreased slightly (Supplementary Fig. 6b). And it features a typical type H₄ hysteresis loop, which is most likely attributed to network effects, due to the presence of mesoporous and microporous structures together[35]. Calculated from the N₂ adsorption isotherm, Brunauer-Emmett-Teller (BET) surface area of CTF-1, CTF-TPA, and CTF-TF are 326.7 m² g⁻¹, 341.2 m² g⁻¹, and 407.7 m² g⁻¹, respectively. As shown in Supplementary Fig. 7, the thermal gravimetric analysis (TGA) results reveal that the CTF samples could be stable up to 400 °C without significant loss of mass. The rapid weight loss only starts from

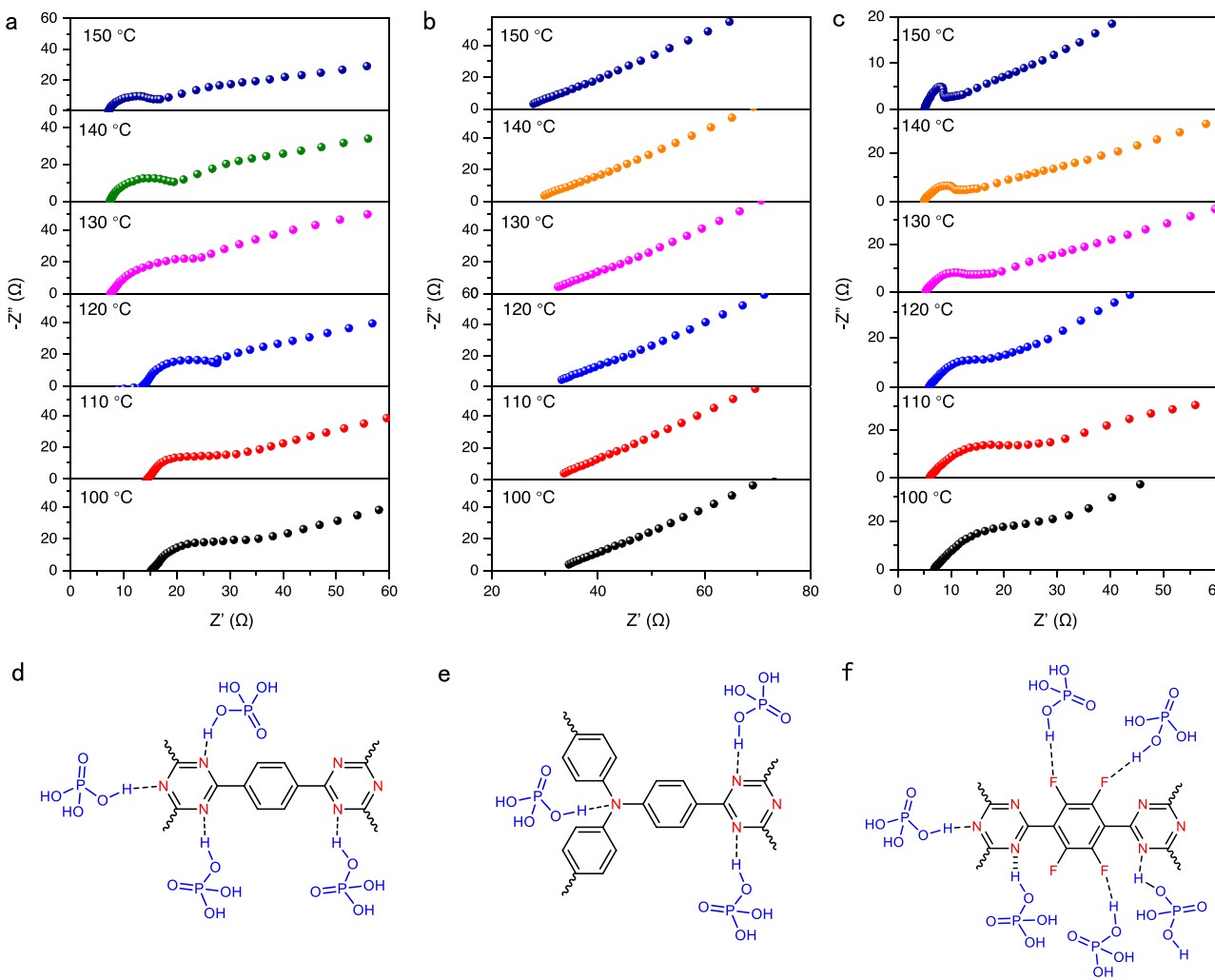

**Fig. 3 | Proton conductivities of CTFs.** Nyquist plots of **a** H₃PO₄@CTF-1; **b** H₃PO₄@CTF-TPA; **c** H₃PO₄@CTF-TF; **d–f** Schematic illustration of the interaction between CTFs and H₃PO₄.

600 °C, implying that the triazine frameworks are highly thermally stable[33,34]. A scanning electron microscope (SEM) was employed to observe the microscopic morphology of this series of polymers. Supplementary Fig. 8 shows that CTFs are stacked nanosheets without special regular morphology. And transmission electron microscope (TEM) show CTF-TF displays more obvious thin layer structure (Supplementary Fig. 9). The above results provide the clear evidences for the successful synthesis of perfluorinated CTF (CTF-TF) via direct cyclotrimerization of aromatic aldehydes using NH₄I as facile nitrogen source.

**Proton conductivities of CTFs loading with H₃PO₄**
With the variable building blocks that can tune the interaction with proton carriers in the present CTFs, we investigated the proton conductivities of the CTFs after being impregnated with phosphoric acid (85%). As shown in Fig. 3, electrochemical impedance spectroscopy (EIS) was tested to evaluate the proton conductivity at different temperatures. The H₃PO₄@CTF-1 exhibits high proton conductivities of $8.34 \times 10^{-2}$, $7.76 \times 10^{-2}$, $6.16 \times 10^{-2}$, $5.62 \times 10^{-2}$, $4.89 \times 10^{-2}$, and $4.32 \times 10^{-2}$ S cm⁻¹ at 150, 140, 130, 120, 110, and 100 °C, respectively. The detailed proton conductivity of CTFs at different temperatures was listed in Supplementary Table 4. Under the same conditions, the proton conductivity of H₃PO₄@CTF-TPA was tested to be $5.54 \times 10^{-2}$ S cm⁻¹ at 150 °C. Remarkably, the conductivity of H₃PO₄@CTF-TF increased up to $1.82 \times 10^{-1}$ S cm⁻¹ at

150 °C, which is much higher than other two CTFs, indicating the electronegative fluorine atoms are important to promote the conductivity. To the best of our knowledge, the proton conductivity of H₃PO₄@CTF-TF is the highest among the reported CTFs and is also comparable to many other porous organic polymers (Supplementary Table 5). For example, the conductivities of the present CTFs can be comparable to H₃PO₄@TPB-DMeTP-COF ($1.91 \times 10^{-1}$ S cm⁻¹ at 160 °C)[5], H₃PO₄@COF-F6 ($4.2 \times 10^{-2}$ S cm⁻¹ at 140 °C)[36] and H₃PO₄@TPB-DABI-COF ($1.52 \times 10^{-1}$ S cm⁻¹ at 160 °C)[37], although they are measured under different conditions.

For the purpose of further highlighting the advantages of our reported new method, F-CTF-CN was synthesized via ionothermal strategy, according to the previous work[14]. Compared to CTF-TF, the F-CTF-CN has a smaller fluorine content of 11.76 wt% and higher surface area of 546.7 m² g⁻¹ (Supplementary Fig. 10), which is ascribed to cleavage of the C−F bonds at high temperature. Loaded with 50% phosphoric acid, the proton conductivity of H₃PO₄@F-CTF-CN was tested to be $6.64 \times 10^{-2}$ S cm⁻¹ at 150 °C (Supplementary Fig. 11), which is much lower than that of H₃PO₄@CTF-TF, and even lower than that of H₃PO₄@CTF−1. This may be due to the lower fluoride content as well as larger particle boundaries[38].

In addition, the proton conductivities of the CTFs are stable, which can be maintained without significant loss after working for 72 h (Supplementary Fig. 12). The activation energy (Ea) can be calculated by using Arrhenius plots to reveal the proton conduction mechanism.

The activation energy of $H_3PO_4$@CTF-1, $H_3PO_4$@CTF-TPA, and $H_3PO_4$@CTF-TF are 0.22, 0.10, and 0.37 eV, respectively (Supplementary Fig. 12). The Ea of the CTFs were all below 0.4 eV, suggesting that the proton conduction processes follow the Grotthuss-type hopping mechanism[39–41]. The Ea is related to proton transfer pathways. However, it is found that the activation energies are not directly correlated to the proton conductivities. We performed the XPS measurements for the three $H_3PO_4$@CTFs and found that the high-resolution P2$p$ spectrums were curve-fitted into two peaks at around134.5 and 135.3 eV assigned to $H_2PO_4^-$ and $H_3PO_4$, respectively[37,42] (Supplementary Fig. 13). The peak area ratios of $H_2PO_4^-$ were calculated, which are in the order of $H_3PO_4$@CTF-TPA (62.4%) > $H_3PO_4$@CTF−1(61.0%) > $H_3PO_4$@CTF-TF (56.3%). Because the energy barriers for proton transfer pathways via $H_3PO_4 \rightarrow H_2PO_4^-$ is the lowest in $H_3PO_4$ loaded proton conducting polymers[43,44], the activation energy may be correlated to the $H_2PO_4^-$ proportion. These results suggest that the higher the proportion of $H_2PO_4^-$, the lower the activation energy it may require for proton transfer.

To probe the reason for the excellent performance of CTF-TF in proton conductivities, we studied the interaction between the CTFs and the $H_3PO_4$ by XPS (Supplementary Fig. 4). After $H_3PO_4$ impregnation, the obvious peaks at around 400.9 eV in the deconvoluted N 1$s$ spectra of $H_3PO_4$@CTF-1 (400.9 eV), $H_3PO_4$@CTF-TPA (400.8 eV) and $H_3PO_4$@CTF-TF (400.9 eV) could be observed, which indicates the formation of protonated pyridine nitrogen[37,45]. The change in the binding energy indicates that the $H_3PO_4$ is locked into CTFs through strong interaction between the alkaline nitrogen and the $H_3PO_4$. Specifically, there is a red-shift of F 1$s$ peak (from 687.1 to 687.7 eV) observed in $H_3PO_4$@CTF-TF (Supplementary Fig. 4d), indicating the electronegative fluorine binding sites in CTF-TF interact with $H_3PO_4$ (Fig. 3d–f)[4,36]. Furthermore, the strength of the hydrogen bonding interactions can be probed by the values of binding energies from theoretical calculation. The $H_3PO_4$ molecules exhibit strong hydrogen bonding interaction when binding to the triazine nitrogen atoms of all three CTFs (Supplementary Fig. 14 and Supplementary Table 6). Notably, getting benefit from the strong electron-withdrawing effect of fluorine, the nitrogen of triazine rings in CTF-TF exhibit stronger binding energy with $H_3PO_4$ as compared to that of CTF-1. In addition, as anchoring points, the fluorine atoms can not only provide more interaction sites, but also facilitate the proton dissociation between CTF-TF and $H_3PO_4$ due to F $\bullet\bullet\bullet$ H − O hydrogen bonds. The binding energy of triazine nitrogen to $H_3PO_4$ in CTF-TPA is the highest among them, but the amount of triazine nitrogen acceptors in the framework structures is halved as compared to that of CTF-1 and CTF-TF. Overall, the CTF-TF gives the highest strength of interaction with phosphoric acid among the three CTFs. The additional hydrogen-bonding interaction sites given by F benefits for the phosphoric acid confinement and proton dissociation in the channels leads to the best proton conductivity of CTF-TF among the series.

The previous researches have shown that the phosphate anion ($H_2PO_4^-$) dynamics favors the long-range proton transport[46,47]. As depicted in Supplementary Fig. 15, the electrostatic potentials (ESP) of the three CTFs exhibit distinct variations. The presence of fluorine (F) atoms in CTF-TF renders its pores electronegative, as indicated by a maximum ESP value of -0.533 eV. Conversely, the pores of CTF-1 exhibit electropositive characteristics, with an electronegative region observed near the nitrogen atoms of the triazine moieties. CTF-TPA adopts a non-planar structure due to the incorporation of triphenylamine units, and an electronegative region is also observed at the N atoms of the triazine moieties. Consequently, CTF-TF exhibits a pronounced electron-withdrawing capacity within its pores, while CTF-1 displays the opposite behavior. CTF-TPA, on the other hand, exhibits electron-withdrawing or electron-donating capabilities at different positions within its pores, depending on the local environment. $H_2PO_4^-$ dynamics that benefit for the proton conduction would be accelerated

in CTF-TF due to electrostatic repulsion, while it may be inhibited in CTF-1 and CTF-TPA owing to the ion-dipole interactions. Therefore, although the activation energy of $H_3PO_4$@CTF-TF is high, the other positive factors still make it show excellent proton conductivity.

We also compared the CTF-TF with the CTF-TF-0.5 that is synthesized by adjusting the proportion of monomers. The surface area of CTF-TF-0.5 is close to that of CTF-TF (401.3 $m^2$ $g^{-1}$ vs. 407.7 $m^2$ $g^{-1}$) (Supplementary Fig. 16), however, the proton conductivity of $H_3PO_4$@CTF-TF-0.5 ($1.22 \times 10^{-1}$ S $cm^{-1}$ at 150 °C, Supplementary Fig. 17), is much lower than that of $H_3PO_4$@CTF-TF, indicating that the fluorine here is a decisive factor for proton conductivity. The reason could be attributed to that the higher content of F atoms in the CTF pores may act as hydrogen-bonding acceptors[48], promoting the formation of hydrogen-bonding networks along the channel and facilitating proton transport[49].

The proton conducting materials possessing good processability and flexibility are desired[50,51]. The research of porous organic polymers in the field of proton conduction is mostly focused on powder form[52–54], and many of the proton conducting membranes of porous organic polymers were only studied under humidity conditions[55–57]. With the present proton conducting CTFs of layered structures in hand, we show that the CTF-1 and CTF-TF can be prepared into nice mixed matrix membranes (MMMs). As shown in Fig. 4a, we adopted commercially available poly(vinylidene fluoride) (PVDF) as the polymer matrix due to its good stability, excellent mechanical strength, and processability. In the ultimate, a high CTF-TF loading of 50 wt% in MMMs can be successfully attained, which is higher than CTF-1 loadings of 40 wt%. This is due to the CTF-TF layers have smaller sizes (Supplementary Fig. 18), which is more conducive for membrane formation[58]. These membranes also show high mechanical strength, allowing them to keep stable without cracks during processing (Supplementary Fig. 19). The tensile test was conducted to evaluate the mechanical properties of MMMs (Supplementary Fig. 20). Stress−strain curves imply that among these membranes, the CTF-TF-40%/PVDF exhibits the highest ultimate stress (20.3 MPa) and Young's modulus (401.26 MPa), which are better than CTF-1-40%/PVDF (ultimate stress: 16.6 MPa, Young's modulus: 386.23 MPa) (Supplementary Table 7). When increasing the CTF-TF contents from 40 wt% to 50 wt%, the mechanical properties of CTF-TF-50%/PVDF could still be maintained to be ultimate stress of 13.8 MPa and Young's modulus of 327.35 MPa. Next, the proton conductivities of these MMMs at high temperature are measured after $H_3PO_4$ loading (Supplementary Table 8). The conductivity values at 150 °C are measured to be $2.07 \times 10^{-2}$ S $cm^{-1}$, $3.68 \times 10^{-2}$ S $cm^{-1}$ and $5.03 \times 10^{-2}$ S $cm^{-1}$ for CTF-1-40%/PVDF, CTF-TF-40%/PVDF, and CTF-TF-50%/PVDF, respectively (Fig. 4 and Supplementary Fig. 21). To our knowledge, the proton conductivity of these membranes still maintained at a high level, comparable to the most reported COF powder samples under the similar conditions (Supplementary Table 5).

## Discussion

This low-temperature method to construct CTFs by a cyclotrimerization reaction using aldehyde monomers and $NH_4I$ as facile nitrogen source provides a synthetic strategy for CTFs under mild conditions. By this gentle approach, the perfluorinated CTF-TF with high fluorine content can be successfully achieved under much milder conditions as compared to the previous reports. Previously, many works have studied the effect of some heteroatoms, such as nitrogen, in the proton conduction at high temperatures, but the effects of other special atoms, such as fluorine atoms, have been rarely explored. In this work, we demonstrate that the fluorinated CTFs are promising for high-temperature proton conduction. It is found that the perfluorinated CTF endowed by the high fluorine content can provide as many anchoring sites as possible to interact with proton carriers and facilitate the proton transport. By using the theoretical DFT calculation, we

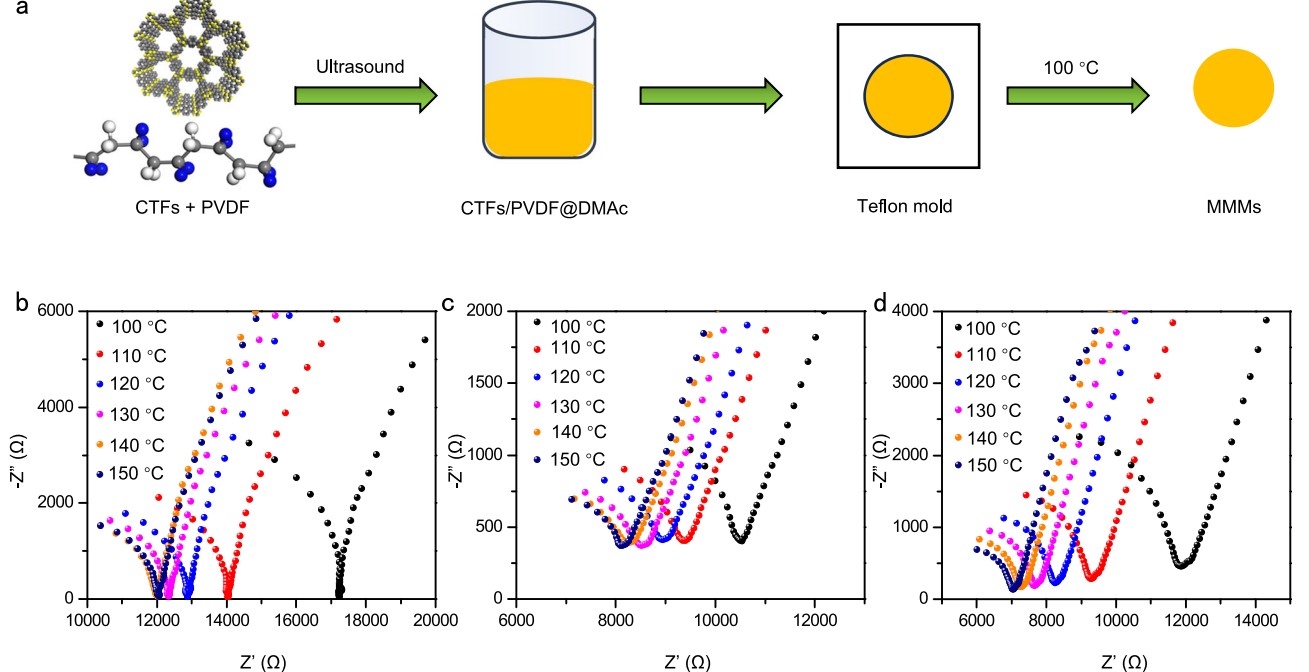

**Fig. 4 | Preparation and proton conductivities of CTFs-based mixed matrix membranes. a** Scheme of the prepare procedure of the mixed matrix membranes (C, gray; N, yellow; H, white; F, blue; DMAc: *N*, *N*-Dimethylacetamide); **b** Nyquist plots of CTF-1-40%/PVDF at different temperatures; **c** Nyquist plots of CTF-TF-40%/PVDF at different temperatures; **d** Nyquist plots of CTF-TF-50%/PVDF at different temperatures.

investigated the interactions between $H_3PO_4$ and CTFs at a molecular level and revealed that the introducing of F atoms can not only directly increase the number of hydrogen acceptors for protons transport, but also enhance the ability of proton dissociation in the proton conduction process.

Based on the unique effect of F atoms, this work further highlights performance of perfluorinated CTF in proton conduction, with a high conductivity up to $1.82 \times 10^{-1}$ S cm$^{-1}$ at 150 °C with $H_3PO_4$. Moreover, the low-temperature derived CTFs can be facilely prepared into mixed matrix membranes that have high proton conductivity (CTF-TF-50%/PVDF: $5.03 \times 10^{-2}$ S cm$^{-1}$ at 150 °C) with good mechanical properties and processability. This work constitutes a facile route for fabricating CTFs with versatile structures as prospective materials for proton conductivity. The precision tuning of the interaction between the framework and proton carriers is vital to control proton conductivities for advancing proton conducting materials.

## Methods
1,4-phthalaldehyde (98%), 2,3,5,6-tetrafluoroterephthalaldehyde (97%), ammonium iodide (NH$_4$I, 99.0%), dimethyl sulfoxide (DMSO, >99%), *N*, *N*-dimethylformamide (DMF, 99.5%), *o*-dichlorobenzene (*o*-DCB, anhydrous, 99%), toluene (99.5%), mesitylene(anhydrous, 98%), copper acetate (Cu(OAc)$_2$, ≥98%), ferric chloride (FeCl$_3$, ≥99.9%), and Fe(NO$_3$)$_3$(AR), Fe(CF$_3$SO$_3$)$_3$ (90%), Fe$_2$(SO$_4$)$_3$ (99%) were purchased from Shanghai Aladdin Biochemical Technology Co, Ltd. Absolute ethanol, and hydrochloric acid (HCl) (37%) were analysis grade and purchased from National Medicines Corporation Ltd. of China. Tris(4-formylphenyl)amine (97%) and hydroxydiacetyl iron, hydrate (Fe(OH)(OAc)$_2$, AR) was Shanghai Macklin Biochemical Technology Co., Ltd.

### Synthesis of CTF-1
CTF-1 was synthesized by a simple polycondensation route. 1,4-phtha-laldehyde (134 mg, 1.0 mmol), NH$_4$I (290 mg, 2.0 mmol) and Fe(OH)(OAc)$_2$ (85 mg, 0.45 mmol) were added to a solution of *o*-DCB (10 mL) in 25 mL round-bottom flask. The mixture was heated, with magnetic

stirring, at 40 °C for 24 h, 80 °C for 24 h, 120 °C for 24 h and then 160 °C for 24 h. After cooling to room temperature, the solid was obtained by filtration, and then the filter cake was washed with DMF, dilute hydro-chloric acid, water, and absolute ethanol for several times, and then vacuum drying at 100 °C for 24 h to obtain a yellow powder sample. (116 mg, Yield: 87.8%).

### Synthesis of CTF-TPA
Tris(4-formylphenyl)amine (165 mg, 0.50 mmol), NH$_4$I (290 mg, 2.0 mmol) and Fe(OH)(OAc)$_2$ (56 mg, 0.3 mmol) were added to a solution of *o*-DCB (8 mL). The mixture was heated, with magnetic stirring, at 40 °C for 24 h, 80 °C for 24 h, 120 °C for 24 h and then 160 °C for 24 h. After cooling to room temperature, the solid was obtained by filtration, and then the filter cake was washed with DMF, dilute hydrochloric acid, water, and absolute ethanol for several times, and then vacuum drying at 100 °C for 24 h to obtain a yellow-green powder sample. (122 mg, Yield: 74.8%).

### Synthesis of CTF-TF
2,3,5,6-tetrafluoroterephthalaldehyde (103 mg, 0.50 mmol), NH$_4$I (150 mg, 1.0 mmol), and Fe(OH)(OAc)$_2$ (45 mg, 0.25 mmol) were added to a solution of o-DCB(8 mL) in 25 mL round-bottom flask. The mixture was heated, with magnetic stirring, at 40 °C for 24 h, 80 °C for 24 h, 120 °C for 24 h and then 160 °C for 24 h. After cooling to room temperature, the solid was obtained by filtration, and then the filter cake was washed with DMF, dilute hydrochloric acid, water, and absolute ethanol for several times, and then vacuum drying at 100 °C for 24 h to obtain a khaki powder sample. (85 mg, Yield: 82.5%).

### Synthesis of CTF-TF-0.5
1,4-phthalaldehyde (67 mg, 0.5 mmol), 2,3,5,6-tetra-fluoroterephthalaldehyde (103 mg, 0.50 mmol), NH$_4$I (290 mg, 2.0 mmol), and Fe(OH)(OAc)$_2$ (85 mg, 0.45 mmol) were added to a solution of *o*-DCB (10 mL) in 25 mL round-bottom flask. The mixture was heated, with magnetic stirring, at 40 °C for 24 h, 80 °C for 24 h,

120 °C for 24 h and then 160 °C for 24 h. After cooling to room temperature, the solid was obtained by filtration, and then the filter cake was washed with DMF, dilute hydrochloric acid, water, and absolute ethanol for several times, and then vacuum drying at 100 °C for 24 h to obtain a yellow powder sample. (134 mg, Yield: 78.8%).

## Synthesis of F-CTF-TF-CN
Tetrafluoroterephthalonitrile (300 mg, 1.5 mmol) and dried $ZnCl_2$ (612 mg, 4.5 mmol) were well mixed and transferred into a Pyrex tube (10 mL). The tube was degassed, sealed, and heated at 400 °C for 48 h. After cooling to room temperature, the product was washed thoroughly with dilute hydrochloric acid, water, and absolute ethanol for several times, and then vacuum drying at 100 °C for 24 h to obtain a black powder sample. (235 mg, Yield: 78.3%).

## Model reaction
Model reaction was performed according to the literature with appropriate modifications[21]. Benzaldehyde (212 mg, 2.0 mmol), $NH_4I$ (290 mg, 2.0 mmol) and $Fe(OH)(OAc)_2$ (85 mg, 0.45 mmol) were added to a solution of o-DCB (10 mL) in 25 mL round-bottom flask. The mixture was heated, with magnetic stirring, at 40 °C for 12 h, 80 °C for 12 h, 120 °C for 12 h and then 160 °C for 12 h. After cooling to room temperature, the solution was diluted with ethyl acetate (20 mL), washed with water (10 mL), extracted with ethyl acetate (3 × 15 mL), dried over anhydrous $Na_2SO_4$ and concentrated in vacuo. The crude product was purified by column chromatography on silica gel (petroleum ether/ethyl acetate: 4/1) to afford the white product. (183 mg, Yield: 86.3%). $^1$H NMR (400 MHz, $CDCl_3$) δ 8.80 – 8.77 (dd, 6H), 7.64 – 7.55 (m, 9H).

## Preparation of H₃PO₄@CTFs
Taking CTF-1 as an example[6]. Firstly, CTF-1 was activated at 180 °C under vacuum for 12 h, and then the commercial phosphoric acid (85 %) was diluted with methanol by a volume ratio of 1:1. Secondly, 150 mg of CTF-1 was added into 5 mL of diluted phosphoric acid, and stirred for 24 h at 25 °C. Finally, the product was filtered and rinse the cake with 10 ml of water, vacuum drying at 100 °C for 24 h to yield about 200 mg of $H_3PO_4$@CTF-1-$H_3PO_4$ with the 33.4% $H_3PO_4$ loading.

$H_3PO_4$@CTF-TPA and $H_3PO_4$@CTF-TF were prepared by following the same procedure. The content of $H_3PO_4$ in CTFs: 27.5%(CTF-TPA-$H_3PO_4$); 46.5%(CTF-TF-$H_3PO_4$). The amounts of $H_3PO_4$ in $H_3PO_4$@CTF-1, $H_3PO_4$@CTF-TPA and $H_3PO_4$@CTF-TF are about 50.2 mg, 41.0 mg, and 69.5 mg, respectively.

## Fabrication of mixed matrix membranes
Taking CTF-TF-50%/PVDF as an example, CTF-TF powder (30 mg) was mixed with PVDF (30 mg) in N, N-Dimethylacetamide (DMAc) (1.0 mL). After sonication for 2 h, the mixture was drop casted onto a teflon mold (diameter, 3 cm), and heated to a temperature of 100 °C in a conventional oven. Then, the solvent was allowed to evaporate at the same temperature for approximately 6 h. The resulting CTF-TF-50%/PVDF was detached from the substrate.

## H₃PO₄ doping of mixed matrix membranes
$H_3PO_4$ doping of mixed matrix membranes were carried out according to the literature with appropriate modifications[59]. Membranes were immersed into 85% $H_3PO_4$ solution at 120 °C for 24 h, and then wiped with filter papers, and dried at 120 °C for 24 h. The weight was calculated based on the difference of the membrane after and before the $H_3PO_4$ doping treatment. The content of $H_3PO_4$: 237 wt% (CTF-1-40%/PVDF); 195 wt% (CTF-TF-40%/PVDF); 247 wt% (CTF-TF-50%/PVDF).

## Characterization
FT-IR spectra were recorded using a Mattson Alpha-Centauri spectrometer (Nicolet iS50, Thermo Scientific). 13 C solid-state spectra were performed on JNM-ECZ600R spectrometer. XPS spectra were acquired on a Kratos Analytical Axis Ultra DLD instrument. Powder X-ray diffraction (PXRD) was performed on a using X-ray diffraction (D8 Advance, Bruker), which was equipped with Cu Kα radiation ($\lambda$ = 1.54056 Å). TGA was performed using a STA 449F5, Perkin Elmer Instruments under $N_2$ atmosphere from 35 °C to 800 °C. Nitrogen absorption and desorption isotherms at 77 K were obtained using Micrometer ASAP 2020 and BSD-16 analyzer after degas in a vacuum at 180 °C for 12 h. Specific surface areas were calculated by using the Brunauer-Emmett-Teller (BET) method. The corresponding pore size distributions were estimated through nonlocal density functional theory (NLDFT). The pore size distributions of CTF-TPA samples were calculated by BJH method. Scanning electron microscopy (SEM) images were obtained with a MAIA3 LMH. Transmission electron microscopy (TEM) images were characterized by transmission electron microscope (Talos L120C G2). Mechanical strength was characterized by a tensile test machine (SHIMADZU AGS-X; 100 N load cell) with a tensile rate of 2 mm min⁻¹.

## Proton conductivity measurement
Powdered samples of $H_3PO_4$@CTFs were filled into a quartz glass tube with a diameter of 5 mm and length of ~3 mm and then compressed between two electrodes. The proton conductivity (σ, S cm⁻¹) of $H_3PO_4$@CTFs were characterized by EIS under the anhydrous condition. AC impedance spectroscopy was measured by CHI600E electrochemical workstation. Temperature-dependent proton conductivities were collected from 100 to 150 °C under the dried air in the closed oven. Prior to the conductivity measurement, the samples were efficiently dried to remove any residual water under 150 °C for 2 hours. The proton conductivities of membranes were measured using a two-electrode system were tested by following the same procedure. The proton conductivity can be calculated by the following equation:

$$\sigma = \frac{L}{RA}$$

where L is the distance between two platinum electrodes (cm). For powder samples, L is ~3 mm; For membrane samples, L is ~1 cm. R is the resistance at a given temperature. A is the cross-section area of the samples (cm⁻²).

The activation energy (Ea) was obtained from the following equation:

$$\ln \sigma = \frac{-Ea}{kT} + \ln \sigma_0$$

where $\sigma$ and $\sigma_0$ denote conductivity and the pre-exponential factor, respectively. T indicates the absolute temperature in kelvin, Ea indicates the activation energy and k denotes the Boltzmann constant.

## Computational details
To depict the electrostatic potential (ESP) profiles of three CTFs and investigate the intermolecular interactions between CTFs and the PA molecule, density functional theory (DFT) calculations were performed using the ORCA quantum-chemistry program package[60]. The geometries of the three CTFs with single-pore structures and the PA molecule within the CTF pores were optimized at the B3LYP-D3(BJ)/TZVP level of theory. Electrostatic potential molecular surfaces were generated using Multiwfn[61] and VMD[62] software. The independent gradient model based on Hirshfeld partition (IGMH) was employed for analysis intermolecular interaction using the same software. Proton dissociation enthalpies (PDE), and binding energies (BE) were also calculated at the B3LYP-D3(BJ)/TZVP level of theory.

## Data availability

All data generated and analyzed in this study are included in the paper and its Supplementary Information, and are also available from authors upon request. Source data are provided with this paper.

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

## Acknowledgements

We greatly acknowledge the financial support from National Natural Science Foundation of China (no. 22275143, 21875078), Key Project of Natural Science Basic Research Program of Shaanxi (no. 2023JC-XJ-14), and Qin Chuangyuan cited high level innovation and entrepreneurship talent project (no. QCYYRCXM–2022–23). S.D. was supported by the Division of Chemical Sciences, Geosciences, and Biosciences, Office of Basic Energy Sciences, US Department of Energy. We appreciate for the support in the characterizations from the Analysis and Testing Center of the Xi'an Jiaotong University.

## Author contributions

Conceiving the project and preparing the materials: S.J. and L.G. Conducting the related analysis.: Q.Z., J.Z. and Z.G. Theoretical calculation and analysis: Z.G. and L.G. Characterization of membranes: C.C. Writing-review and editing: S.J., S.W., X.Z. and S.D. All authors discussed the results and commented on the manuscript.

## Competing interests

All authors declare no competing interests.
