## [Peer review file · Nature Communications]

REVIEWER COMMENTS

Reviewer #1 (Remarks to the Author):

The authors of this manuscript have synthesized a perfluorinated covalent triazine framework (CTF) by a low-temperature method in open-air atmosphere. This CTF has a high proton conductivity, which is important for renewable and hydrogen energy applications. The authors have confirmed the structure of the CTFs by various characterization techniques and have shown that the incorporation of fluorine atoms into the CTFs increased the proton conductivity by enhancing the interaction with H₃PO₄, which was used as a proton carrier. However, the mechanism of how fluorine atoms affect the proton conductivity is not well illustrated in the manuscript, and the evidence provided by the authors is insufficient to support their claim. Therefore, I suggest that this manuscript can be accepted by Nature Communications only after the following questions are properly addressed.

1. The authors should compare the fluorine content of their CTF-TF with other CTFs that have been prepared by different methods and reported in the references. This would show how effective their synthesis method is for introducing fluorine atoms into the CTFs.
2. The authors should also compare the proton conductivity of their CTF-TF with other fluorinated CTFs that have been reported in the references. This would show how significant their improvement in proton conductivity is compared to previous studies.
3. The authors have suggested that the strong interaction between CTFs and H₃PO₄ is important for achieving high proton conductivity. They have used XPS measurements to show that the incorporation of fluorine atoms into the CTFs enhanced the interaction between CTF-TF and H₃PO₄, leading to higher proton conductivity. However, they have not provided a clear correlation between the strength of the interaction and the proton conductivity for the three CTFs that they have synthesized. It is suggested that they quantify the interaction between CTFs and H₃PO₄ and provide a correlation between the interaction strength and the proton conductivity for each CTF.
4. The authors should also consider the effect of BET surface area on the proton conductivity of their CTFs. The three CTFs that they have synthesized have different BET surface areas, which may influence their proton conductivity. The authors should either normalize their proton conductivity data to the BET surface area or use CTFs with similar surface areas to exclude this effect and demonstrate that fluorine incorporation is indeed responsible for enhancing the proton conductivity.
5. The authors should explain why there is no clear correlation between the activation energy and the proton conductivity for their CTFs. They have reported that H₃PO₄@CTF-1, H₃PO₄@CTF-TPA and H₃PO₄@CTF-TF have activation energies of 0.22, 0.10 and 0.37 eV, respectively, but their proton conductivities do not correlate with this order. They should provide some possible reasons for this discrepancy and discuss how it affects their interpretation of the results.
6. It is suggested that the authors use DFT or FTIR to investigate the interaction between H₃PO₄ and CTFs at a molecular level and further elucidate the mechanism of how fluorine atoms enhance the proton conductivity. This would provide more insight into the role of fluorine atoms in modifying the electronic structure and polarity of the CTFs and how they affect the mobility and transfer of protons.

Reviewer #2 (Remarks to the Author):

Guan et al. present an interesting paper on the synthesis of fluorinated triazine framework that show good proton conductivity when impregnated with phosphoric acid. The work may generally be publishable but lacks the high degree of novelty and significance needed for Nature Communications. In addition, several issues are unclear (see below). Therefore, I do not recommend publication in this journal.

The impedance measurements lack information about the relative humidity. Proton conductivity is known to be strongly dependent on humidity, including in organic polymers such as these. A careful control over and systematic variation of humidity would be mandatory to assess how much the

conductivity is mediated by water.

What does "semi-crystalline" mean (XRD)? The term needs explanation. In addition, a peak in the XRD diagram (Suppl. fig. 5) at ca. 18° is not explained, either.

The interpretation of the N₂ physisorption analysis is questionable. Hysteresis closure at ca. $p/p_0 = 0.42$ is most likely due to network effects ('forced hysteresis closure', 'tensile strength effect') rather than caused by mesoporosity.

According to the TGA data, the materials not really seem to be stable at temperature above ca. 150 °C.

Reviewer #3 (Remarks to the Author):

The manuscript by S. Jin and workers describes the low-temperature synthesis of covalent triazine frameworks and investigation of their proton conductivities. The synthesis approach is a good alternative to those reported procedures and the proton conducting performances of the obtained CTFs are high. I would like to recommend acceptance for publication after addressing the following points.

1 The authors should emphasize the novelty and advantage of the NH₄I-involved procedure and provide the comparison with other processes.

2 The proton conductivity results should be explained considering the chemical structures.

3 "The activation energy of H₃PO₄@CTF-1, H₃PO₄@CTF-TPA and H₃PO₄@CTF-TF are 0.22, 0.10 and 0.37 eV, respectively."

Please explain the reason behind the difference in activation energy.

4 Section "Discussion" is not discussion, is the summary.

5 The optimum conditions for the synthesis of CTFs are not given.

The yield provided in the text should be specified for some reaction temperature.

All the yield results should be provided for every reaction temperatures.

6 The so-called host-guest interaction in CTFs, the authors should provide some more convinced evidence, experimental or theoretical results.

7 Some typo and grammar errors should be corrected. The following are some examples.

the frameworks with more strengthened anchoring sites is (are) preferred

130 ppm ~145 ppm (130 ~145 ppm)

"Scanning electron microscope (SEM) and transmission electron microscope (TEM) were employed to observe the microscopic morphology"

Only the SEM results are employed to observe the morphology.

The proton conducting materials possessing good processability and flexibility is (are) desired.

The research of porous organic polymers in the field of proton conduction are (is) mostly focused on powder form

interreact (interact) with H_3PO_4

Responses to Reviewer #1:

[Remarks to the Author]: The authors of this manuscript have synthesized a perfluorinated covalent triazine framework (CTF) by a low-temperature method in open-air atmosphere. This CTF has a high proton conductivity, which is important for renewable and hydrogen energy applications. The authors have confirmed the structure of the CTFs by various characterization techniques and have shown that the incorporation of fluorine atoms into the CTFs increased the proton conductivity by enhancing the interaction with H₃PO₄, which was used as a proton carrier. However, the mechanism of how fluorine atoms affect the proton conductivity is not well illustrated in the manuscript, and the evidence provided by the authors is insufficient to support their claim. Therefore, I suggest that this manuscript can be accepted by Nature Communications only after the following questions are properly addressed.

Response: We are grateful for your constructive suggestions and giving us an opportunity to revise this manuscript. We have carefully revised the manuscript considering the thoughtful suggestions. In particular, we provide more convinced evidences, experimental and theoretical results, for host-guest interaction and have further discussed the role of fluorine atoms. The details of our responses to the comments were listed below.

[1] The authors should compare the fluorine content of their CTF-TF with other CTFs that have been prepared by different methods and reported in the references. This would show how effective their synthesis method is for introducing fluorine atoms into the CTFs.

Response: Thank you for your insightful comments. Following your suggestion, we listed the fluorinated CTFs reported previously and made the detailed comparison, which shows the clear advantages of our method (see Supplementary Table 1). From this Table, the advantages of this work in synthesizing perfluorinated CTFs are illustrated from the following aspects:

Firstly, we compared the fluorine content of CTF-TF with the CTFs reported in references. It shows that the fluorine content in our method is higher than most reported CTFs synthesized by ionothermal method at high temperatures (>400 °C), due to the cleavage of C-F bond at high temperatures (>400 °C). There is only one case with fluorine content of 23.8 wt% for fluorinated CTFs via conventional ionothermal method at high temperatures (>400 °C) (*Energy Environ. Sci.* **2013**, 6, 3684-3692), and the fluorine contents of most reported CTFs are lesser than 15 wt% (*Chem. Eng. J.* 2020, 400, 125967; *Sep. Purif. Technol.* **2022**, 290, 120857). While, in the fluorinated CTF obtained by strong acid catalysis, the fluorine content was not explicitly

disclosed (*J. Am. Chem. Soc.* **2020**, 142, 6856–6860), where the synthesis is still conducted under high temperatures (250-350 °C) and particularly the surface area is only 2 m² g⁻¹, much lower than that of CTF-TF in our work (407.7 m² g⁻¹), which is not preferable to accommodate more proton carriers.

Secondly, to achieve excellent proton conductivity, it is important to retain intact of the chemical structure during synthesis and high content of the hydrogen bond acceptors in the framework, so it is highly preferable to synthesize the materials at mild conditions. To achieve fluorinated CTFs with higher fluorine content, the researchers have struggled to developed new methods to synthesize them at lower temperatures. Although an improved ionothermal method to synthesize a perfluorinated CTF at relatively lower temperatures has been reported (*Angew. Chem. Int. Ed.* **2021**, 60, 25688–2569), which exhibits a fluorine content (31wt%) comparable to that of CTF-TF (30.2wt%) in this work, the operation of synthesis under vacuum system and rather higher temperature (275 °C) are required and the catalyst needs to be synthesized in much complex procedures. In contrast, the method reported in this work can be conducted in the air atmosphere, and requires much lower temperatures (<160 °C). Also, the catalyst we used in this method is cheap and easily commercially available. So, this method is much milder than the reported methods, making it more accessible to the high fluorinated CTF and also avoiding of damage by the high temperature to the chemical structure of the frameworks.

To make it more clearly, the Supplementary Table 1 and the corresponding discussion are added in the revised manuscript as follows:

Supplementary Table 1. Comparison of previous reported fluorinated CTFs prepared by different methods with this work.

Monomer	Methods	Conditions	F contents (wt %)	Surface area (m ² g ⁻¹)	References
	NH ₄ I-involved method	Fe(OAc) ₃ ; < 160°C; Open system	30.2	407.7	This Work
	Ionothermal strategy	ZnCl ₂ , 400 °C	23.8	623	Energy Environ. Sci. 2013, 6 , 3684-369
	Ionothermal strategy	ZnCl ₂ , 600°C	3.7	1558	J. Mater. Chem. A , 2019, 7 , 17277-17282

	Superacid-catalysis	CF ₃ SO ₃ H, 250 °C, sealed system; 350 °C, N ₂	--	2 (almost nonporous)	J. Am. Chem. Soc. 2020, 142 , 6856–6860
	ionothermal strategy	[Zn(NTf ₂) ₂], 275 °C, sealed system	31	367	Angew. Chem. Int. Ed. 2021, 60 , 25688–256
	ionothermal strategy	ZnCl ₂ , 400 °C	3.3	638	Sep. Purif. Technol. 2022, 290 , 120857

“The new synthetic approach enables the synthesis of perfluorinated CTF with high fluorine content and surface area under much milder conditions as compared to the reported methods (Supplementary Fig. 1 and Supplementary Table 1) ^{14,17-20}.”

“Obviously, the fluorine content of CTF-TF is comparable to the highest values reported so far and is much higher than those of most fluorinated CTFs (Supplementary Table 1). Notably, this marks the first instance of synthesizing a type of perfluorinated CTF with a significantly high fluorine content through such gentle methods. Previous conventional approaches struggled to achieve this level of perfluorination under mild conditions.”

[2] The authors should also compare the proton conductivity of their CTF-TF with other fluorinated CTFs that have been reported in the references. This would show how significant their improvement in proton conductivity is compared to previous studies.

Response: Thank you for your valuable suggestion. Recently, CTF materials have emerged as a promising platform for high temperature proton conduction applications. However, to the best of our knowledge, there is no report using fluorinated CTFs for the proton conduction at high temperatures so far. Thus, this work reports the first example of fluorinated CTFs applied to proton conduction at high temperature.

To directly compare with other methods, we thereby prepared a F-CTF-CN sample via conventional ionothermal strategy at 400 °C according to the literature method (*Energy Environ. Sci.* **2013**, 6, 3684-369). It is found that the surface area of the F-CTF-CN (546.7 m² g⁻¹) is higher than this method, but the fluorine content (11.76 wt%) is much lower than the CTF-TF (30.2wt%) of this work, which is due to the cleavage of the C-F bond at high temperature. After loading with phosphoric acid (50%), the proton conductivity of H₃PO₄@F-CTF-CN was tested

to be only $6.64 \times 10^{-2} \text{ S cm}^{-1}$ at $150 \text{ }^\circ\text{C}$, which is much lower than that of $\text{H}_3\text{PO}_4@\text{CTF-TF}$ ($1.82 \times 10^{-1} \text{ S cm}^{-1}$ at $150 \text{ }^\circ\text{C}$). These results clearly indicate the present method can benefit for obtaining high fluorine content CTF and significantly increase the proton conductivity.

To make it more clearly, the corresponding discussion, supplementary figures and references have been added into the revised manuscript as follows:

“For the purpose of further highlighting the advantages of our reported new method, F-CTF-CN was synthesized via ionothermal strategy, according to the related previous work¹⁴. Compared to CTF-TF, the F-CTF-CN has a smaller fluorine content of 11.76 wt% and higher surface area of $546.7 \text{ m}^2 \text{ g}^{-1}$ (Supplementary Fig. 10), which is ascribed to the cleavage of C–F bonds at high temperature. Loaded with 50% phosphoric acid, the proton conductivity of $\text{H}_3\text{PO}_4@\text{F-CTF-CN}$ was tested to be $6.64 \times 10^{-2} \text{ S cm}^{-1}$ at $150 \text{ }^\circ\text{C}$ (Supplementary Fig. 11), which is much lower than that of $\text{H}_3\text{PO}_4@\text{CTF-TF}$, and even lower than that of $\text{H}_3\text{PO}_4@\text{CTF-1}$. This may be due to the lower fluoride content as well as larger particle boundaries³⁸.”

14. Zhao, Y. F., Yao, K. X., Teng, B. Y., Zhang, T. & Han, Y. A perfluorinated covalent triazine-based framework for highly selective and water-tolerant CO_2 capture. *Energy Environ. Sci.* **6**, 3684-3692 (2013).

38. Zhou, M.-Y. et al. Single-crystal superprotonic conductivity in an interpenetrated hydrogen-bonded quadruplex framework. *Chem. Commun.* **58**, 771-774 (2022).

Supplementary Fig. 10. (a) FT-IR spectra of F-CTF-CN; (b) N_2 adsorption and desorption isotherms (77 K) curves of F-CTF-CN.

Supplementary Fig. 11. Proton conductivity of F-CTF-CN. (a) Nyquist plots of $\text{H}_3\text{PO}_4@\text{F-CTF-CN}$; (b) Arrhenius plots for $\text{H}_3\text{PO}_4@\text{F-CTF-CN}$. The $\text{H}_3\text{PO}_4@\text{F-CTF-CN}$ exhibits proton conductivities of 2.35×10^{-2} , 2.55×10^{-2} , 3.12×10^{-2} , 3.77×10^{-2} , 5.21×10^{-2} , and $6.64 \times 10^{-2} \text{ S cm}^{-1}$ at 100, 110, 120, 130, 140 and $150 \text{ }^\circ\text{C}$, respectively.

[3] The authors have suggested that the strong interaction between CTFs and H_3PO_4 is important for achieving high proton conductivity. They have used XPS measurements to show that the incorporation of fluorine atoms into the CTFs enhanced the interaction between CTF-TF and H_3PO_4 , leading to higher proton conductivity. However, they have not provided a clear correlation between the strength of the interaction and the proton conductivity for the three CTFs that they have synthesized. It is suggested that they quantify the interaction between CTFs and H_3PO_4 and provide a correlation between the interaction strength and the proton conductivity for each CTF.

Response: Thank you very much for the constructive suggestion. According to the Reviewer's comments, we analyzed the chemical structures of CTFs and conducted theoretical calculations to estimate the binding energies between CTFs and H_3PO_4 , which may quantify the strength of interaction between them.

Firstly, the interaction between the CTFs and H_3PO_4 could be evaluated by the density of the interaction sites as hydrogen bond acceptors from the chemical structures of CTFs. In all the three CTFs, the triazine units are the major hydrogen bond acceptors to bind with H_3PO_4 . In addition to the triazine units, we can find that CTF-TF can provide abundant F atoms as additional hydrogen bond acceptors in the pores as compared to CTF-1. The large amount of F atoms renders CTF-TF comprise the largest density of interaction sites with protons in the framework among the three CTFs. The high density of hydrogen bond acceptors in the frameworks can benefit for the formation of hydrogen bonding networks, which facilitates the proton transport in the proton conduction. There are also two types of hydrogen bond acceptors in CTF-TPA, that are, N (triphenylamine) and N (triazine), however, the number of triazine N sites (major hydrogen bond acceptors) is only half of CTF-TF and CTF-1, making it inferior to form hydrogen bonding network and facilitate proton transport in proton conduction.

Secondly, according to the theoretical calculation results (as shown in Supplementary Fig. 14 and Supplementary Table 6), the CTF-TF displays a higher binding energy (-18.676 kcal/mol) and lower proton dissociation enthalpy (4.481 eV) between the triazine N sites and phosphoric acid as compared to CTF-1 (-16.150 kcal/mol and 4.543 eV), clearly indicating that the introducing of F atoms can indeed enhance the interaction between CTFs and phosphoric acid and facilitate the proton dissociation in the proton conduction simultaneously. Especially, the calculation shows that the interaction of phosphoric acid with F atoms has lower proton dissociation enthalpy (4.265 eV) as compared to the N atoms in the CTFs, which further facilitates the proton dissociation in the proton conduction. In the CTF-TPA, despite it exhibits higher binding energy of -22.347 kcal/mol on the N (triazine) sites than other CTFs, its halved

N (triazine) hydrogen bond acceptors in CTF-TPA counteracts the positive effect on the proton conduction as compared to CTF-TF and CTF-1.

According to these analyses, we conclude that the introducing of F atoms can not only make the CTF-TF comprise the maximum number of hydrogen bond acceptors, but also benefit for the proton dissociation and transportation, thereby it effectively enhances the proton conductivity of CTF-TF, and that the overall strength of the interaction between the CTFs and phosphoric acid could be correlated with the proton conductivity performance.

To make it more clearly about the relationship and effect of fluorine atoms, the corresponding statement and discussion are added in the revised manuscript as follows:

“Furthermore, the strength of the hydrogen bonding interactions can be probed by the values of binding energies from theoretical calculation. The H_3PO_4 molecules exhibit strong hydrogen bonding interaction when binding to the triazine nitrogen atoms of all three CTFs (Supplementary Fig. 14). Notably, getting benefit from the strong electron-withdrawing effect of fluorine, the nitrogen of triazine rings in CTF-TF exhibit stronger binding energy with H_3PO_4 as compared to that of CTF-1. In addition, as anchoring points, the fluorine atoms can not only provide more interaction sites, but also facilitate the proton dissociation between CTF-TF and H_3PO_4 due to $F \cdots H-O$ hydrogen bonds. The binding energy of triazine nitrogen to H_3PO_4 in CTF-TPA is the highest among them, but the amount of triazine nitrogen acceptors in the framework structures is halved as compared to that of CTF-1 and CTF-TF. Overall, the CTF-TF gives the highest strength of interaction with phosphoric acid among the three CTFs. The additional host-guest interaction sites given by F benefits for the phosphoric acid confinement and proton dissociation in the channels of CTF-TF, leading to better proton conductivity among the series.”

Supplementary Fig. 14. Graphic representations of the binding energy of H_3PO_4 to various interaction sites for CTFs. (a) CTF-1(Triazine N), (b) CTF-TPA (Triazine N), (c) CTF-TPA (Triphenylamine N), (d) CTF-TF (F) and (e) CTF-TF (Triazine N). (H, white; P, brown; O, red; N, blue; C, cyan; F, LT Magenta).

Supplementary Table 6. Summary of the binding energy and proton dissociation enthalpy results of the theoretical calculation (PA:H₃PO₄).

	CTF-TF+PA (Triazine N...H)	CTF-TF+PA (F...H)	CTF-1+PA (Triazine N...H)	CTF- TPA+PA (Triphenyl- amine N...H)	CTF- TPA+PA (Triazine N...H)
Hirshfeld partition (IGMH)					Proton dissociation enthalpy (eV)	4.481	4.265	4.543	4.515	4.496
Binding energy (kcal/mol)	-18.767	-5.328	-16.150	-13.767	-22.347

[4] The authors should also consider the effect of BET surface area on the proton conductivity of their CTFs. The three CTFs that they have synthesized have different BET surface areas, which may influence their proton conductivity. The authors should either normalize their proton conductivity data to the BET surface area or use CTFs with similar surface areas to exclude this effect and demonstrate that fluorine incorporation is indeed responsible for enhancing the proton conductivity.

Response: Thank you for this very insightful suggestion. To probe the impact of surface area and critical role of fluorine, we prepared a sample of CTF-TF-0.5 with lesser fluorine content by co-polymerizing, in which the amount of fluorine monomer decreased by 50%. The BET surface area of CTF-TF-0.5 (401.3 m² g⁻¹) is very close to that of CTF-TF (407.7 m² g⁻¹), but its fluorine content of 18.92 wt% is obviously smaller than CTF-TF (30.2wt%). It shows that the proton conductivity of H₃PO₄@CTF-TF-0.5 (1.22×10⁻¹ S cm⁻¹ at 150 °C) is much lower than the CTF-TF (1.82×10⁻¹ S cm⁻¹ at 150 °C) under the same conditions, clearly indicating that the fluorine is indeed crucial factor for enhancing the proton conductivity. The reason could be attributed to that the higher content of F atoms may act as hydrogen bond acceptor to stabilize the proton network and facilitate proton conduction.

To make it more clearly, the corresponding explanation with references are added in the

revised manuscript as follows:

“We also compared the CTF-TF with the CTF-TF-0.5 that is synthesized by adjusting the proportion of monomers. The surface area of CTF-TF-0.5 is close to that of CTF-TF ($401.3 \text{ m}^2 \text{ g}^{-1}$ vs. $407.7 \text{ m}^2 \text{ g}^{-1}$) (Supplementary Fig. 16), however, the proton conductivity of $\text{H}_3\text{PO}_4@\text{CTF-TF-0.5}$ ($1.22 \times 10^{-1} \text{ S cm}^{-1}$ at $150 \text{ }^\circ\text{C}$, Supplementary Fig. 17), is much lower than that of $\text{H}_3\text{PO}_4@\text{CTF-TF}$, indicating that the fluorine here is a decisive factor for proton conductivity. The reason could be attributed to that the higher content of F atoms in the CTF pores may act as hydrogen-bonding acceptors⁴⁸, promoting the formation of hydrogen-bonding networks along the channel and facilitating proton transport⁴⁹.”

Supplementary Fig. 16. (a) synthesis of CTF-TF-0.5; (b) FT-IR spectra of CTF-TF-0.5; (c) N_2 adsorption and desorption isotherms (77 K) curves of CTF-TF-0.5.

Supplementary Fig. 17. Proton conductivity of CTF-TF-0.5 (a) Nyquist plots of $\text{H}_3\text{PO}_4@\text{CTF-TF-0.5}$; (b) Arrhenius plots for $\text{H}_3\text{PO}_4@\text{CTF-TF-0.5}$. The $\text{H}_3\text{PO}_4@\text{CTF-TF-0.5}$ exhibits proton conductivities of 4.37×10^{-2} , 5.10×10^{-2} , 5.97×10^{-2} , 6.78×10^{-2} , 8.08×10^{-2} , and $1.22 \times 10^{-1} \text{ S cm}^{-1}$ at 100, 110, 120, 130, 140 and 150 $^\circ\text{C}$, respectively.

48. Liu, Y. et al. Tough, stable and self-healing luminescent perovskite-polymer matrix applicable to all harsh aquatic environments. *Nat. Commun.* **13**, 1338 (2022).

49. Liu, S. et al. Construction of dense H-bond acceptors in the channels of covalent organic frameworks for proton conduction. *J. Mater. Chem. A* **11**, 13965-13970 (2023).

[5] The authors should explain why there is no clear correlation between the activation energy and the proton conductivity for their CTFs. They have reported that H₃PO₄@CTF-1, H₃PO₄@CTF-TPA and H₃PO₄@CTF-TF have activation energies of 0.22, 0.10 and 0.37 eV, respectively, but their proton conductivities do not correlate with this order. They should provide some possible reasons for this discrepancy and discuss how it affects their interpretation of the results.

Response: Thank you for this insightful question. Indeed, there is no clear correlation between the activation energy and the proton conductivity in our experiments. Low activation energies are generally favorable for proton conduction. However, the proton conductivity is not solely related to the activation energy. The relationship between the temperature-dependent proton conductivity (σ) and the activation energy (Ea) follows the Arrhenius equation (Equation 1).

$$\sigma = \sigma_0 \exp(-Ea/kT) \dots\dots \text{Equation 1}$$

where T is the temperature, σ_0 is an experimental prefactor and k is the Boltzmann constant.

This means that the experimental prefactor (σ_0) is also an important influencing factor. In the proton conduction, the experimental prefactor (σ_0) and the activation energy (Ea) also have the following relationship, which is known as Meyer–Neldel rule (MNR) (Equation 2):

$$\ln\sigma_0 = Ea/kT_{iso} + \ln\sigma_{00} \dots\dots \text{Equation 2}$$

where σ_{00} and T_{iso} are constants, which depends on the proton conduction system itself.

Combining Equations (1) and (2), the Arrhenius relationship can be written as

$$\sigma = (\sigma_{00}/T) e^{(-Ea/k)(1/T-1/T_{iso})} \dots\dots \text{Equation 3}$$

Equation 3 shows that, for $T \leq T_{iso}$, the highest values of σ are obtained for the lowest values of Ea , while for $T \geq T_{iso}$, the highest σ corresponds to the largest Ea .

The above equation shows that the change in prefactor compensates the change in activation energy. The conductivity variation of proton conductors follows the Meyer-Neldel rule, which is manifested as a compensation principle, that is, when the activation energy of proton conduction decreases, the prefactor in the conductivity formula will also decrease accordingly, resulting in a weakened optimization effect of the reduced activation energy on conductivity (*Adv. Energy Mater.*, **2022**, 12, 2102939). Thus, according to these principles, the proton conductivity and activity energy are not always correlated to each other clearly.

Furthermore, the factors that influence the proton conductivity can also be described in the following relationship (*Nat. Mater.* **2009**, 8, 831–836; *Chem. Mater.* 2016, 28, 1489–1494):

$$\sigma(T) = \sum n_i q_i \mu_i$$

where n is the number of carriers, q is the charge, and μ is the mobility of the protons.

According to this relationship, the high activation energy may not be favorable for high mobility of the protons (μ), but it only partially affects the proton conduction to some extent. Whereas, the number of the hydrogen bond acceptors in the CTFs could mainly determine the level of the proton conduction, which is due to that the more interaction sites between CTFs and phosphoric acid can not only result in higher number of proton carriers (n), but also form more hydrogen bonding networks to increase the mobility of the protons (μ). Further, the stronger interaction between the CTFs and phosphoric acid also facilitates the mobility of the protons (μ), benefiting for the proton conduction. As is the case in this work, the CTF-TF has the largest density of anchoring and interaction sites (F+N-triazine) with phosphoric acid than the CTF-1 and CTF-TPA, thereby leading to the highest proton conductivity. In contrast, even though the CTF-TPA has the lowest activation energy, the least density of effective hydrogen bond acceptors (triazine N sites) in the framework makes it exhibit the lowest proton conductivity.

The activation energy should be related to the proton transport pathways, which affects the energy barrier for proton conduction. We further probed the reason for the discrepancy of activation energy by experimental analysis and discussed the correlation with the proton conductivity. The corresponding statements and Supplementary Figures for discussion of the possible reasons for this discrepancy are added in the revised manuscript as follows:

“The E_a is related to proton transfer pathways. However, it is found that the activation energies are not directly correlated to the proton conductivities. We performed the XPS measurements for the three $H_3PO_4@CTFs$ and found that the high-resolution P 2p spectrums could be curve-fitted into two peaks at around 134.5 and 135.3 eV assigned to $H_2PO_4^-$ and H_3PO_4 , respectively^{37,42} (Supplementary Fig. 13). The peak area ratios of $H_2PO_4^-$ were calculated, which are in the order of $H_3PO_4@CTF-TPA$ (62.4%) > $H_3PO_4@CTF-1$ (61.0%) > $H_3PO_4@CTF-TF$ (56.3%). Because the energy barriers for proton transfer pathways via $H_3PO_4 \rightarrow H_2PO_4^-$ is the lowest in H_3PO_4 loaded proton conducting polymers^{43, 44}, the activation energy can be correlated to the $H_2PO_4^-$ proportion. These results suggest that the higher the proportion of $H_2PO_4^-$, the lower the activation energy it may require for proton transfer.”

“The previous researches have shown that that the phosphate anion ($H_2PO_4^-$) dynamics favors the long-range proton transport^{46,47}. As depicted in Supplementary Fig. 15, the electrostatic potentials (ESP) of the three CTFs exhibit distinct variations. The presence of fluorine (F) atoms in CTF-TF renders its pores electronegative, as indicated by a maximum

ESP value of -0.533 eV. Conversely, the pores of CTF-1 exhibit electropositive characteristics, with an electronegative region observed near the nitrogen atoms of the triazine moieties. CTF-TPA adopts a non-planar structure due to the incorporation of triphenylamine units, and an electronegative region is also observed at the N atoms of the triazine moieties. Consequently, CTF-TF exhibits a pronounced electron-withdrawing capacity within its pores, while CTF-1 displays the opposite behavior. CTF-TPA, on the other hand, exhibits electron-withdrawing or electron-donating capabilities at different positions within its pores, depending on the local environment. H_2PO_4^- dynamics that benefit for the proton conduction would be accelerated in CTF-TF due to electrostatic repulsion, while it may be inhibited in CTF-1 and CTF-TPA. Therefore, although the activation energy of H_3PO_4 @CTF-TF is high, the other positive factors still make it result in excellent proton conductivity.”

The related references and Figures are added in the revised manuscript as follows:

Supplementary Fig. 13. High-resolution P 2p XPS spectra of (a) H_3PO_4 @CTF-1, (b) H_3PO_4 @CTF-TPA, and (c) H_3PO_4 @CTF-TF.

Supplementary Fig. 15. Electrostatic potential of (a) CTF-1, (b) CTF-TPA, and (c) CTF-TF.

37. Li, J., Wang, J., Wu, Z. Z., Tao, S. S. & Jiang, D. L. Ultrafast and stable proton conduction in polybenzimidazole covalent organic frameworks via confinement and activation. *Angew. Chem. Int. Ed.* **60**, 12918-12923 (2021).

42. Jiang, G. et al. Tuning the interlayer interactions of 2D covalent organic frameworks enables an ultrastable platform for anhydrous proton transport. *Angew. Chem. Int. Ed.* **61**, e202208086 (2022).

43. Ma, Y. L., Wainright, J. S., Litt, M. H. & Savinell, R. F. Conductivity of PBI membranes for high-temperature polymer electrolyte fuel cells. *J. Electrochem. Soc.* **151**, A8-A16 (2004).

44. Li, S., Fried, J. R., Sauer, J., Colebrook, J. & Dudis, D. S. Computational chemistry and molecular simulations of phosphoric acid. *Int. J. Quantum. Chem.* **111**, 3212-3229 (2011).

46. Traer, J. W., Britten, J. F. & Goward, G. R. A solid-state NMR study of hydrogen-bonding networks and ion dynamics in benzimidazole salts. *J. Phys. Chem. B* **111**, 5602-5609 (2007).

47. Asensio, J. A., Sánchez, E. M. & Gómez-Romero, P. Proton-conducting membranes based on benzimidazole polymers for high-temperature PEM fuel cells. A chemical quest. *Chem. Soc. Rev.* **39**, 3210–3239 (2010).

[6] It is suggested that the authors use DFT or FTIR to investigate the interaction between H₃PO₄ and CTFs at a molecular level and further elucidate the mechanism of how fluorine atoms enhance the proton conductivity. This would provide more insight into the role of fluorine atoms in modifying the electronic structure and polarity of the CTFs and how they affect the mobility and transfer of protons.

Response: Thank you for the insightful comments. Following the Reviewer's suggestion, we conducted the theoretical DFT calculation to probe the interaction between H₃PO₄ and CTFs. According to the calculation results, the CTF-TF exhibits higher binding energy (-18.676 kcal/mol) and lower proton dissociation energy (4.481 eV) on the N (triazine) sites as compared to CTF-1 (-16.150 kcal/mol and 4.543 eV), indicating the introducing of F atoms can change the polarity of the framework and indeed enhance the interaction between the CTFs and H₃PO₄. Especially, the calculation shows that the F atoms have much lower proton dissociation energy (4.265 eV) as compared to the N atoms in the CTFs series, which further facilitates the proton dissociation and transport in the proton conduction. As for CTF-TPA, as compared to CTF-TF and CTF-1, it exhibits a higher binding energy of -22.347 kcal/mol with H₃PO₄ on the N (triazine) sites, but a lower binding energy on triphenylamine N sites. Despite its higher binding energy on the N (triazine) sites, the number of N (triazine) sites is only half of that in CTF-TF and CTF-1, which counteracts the positive effect of binding energy on the proton conduction. Thus, the introducing of F atoms can not only directly increase the density of hydrogen bond

acceptors for higher proton number, but also enhance the ability of proton dissociation and transport in the proton conduction process.

Moreover, to demonstrate the electronic structure and polarity of the CTFs, we drew the electrostatic potential (ESP) maps of CTFs, which shows that the presence of F atoms in CTF-TF renders its pores electronegative, as indicated by a highest ESP value of -0.533 eV. Proton transfer usually tends to occur between H_3PO_4 and H_2PO_4^- (*J. Electrochem. Soc.*, **2004**, 151, A8–A16; *Int. J. Quantum. Chem.*, **2011**, 111, 3212–3229). Therefore, phosphate anion (H_2PO_4^-) dynamics is important to contribute to long-range proton transport (*J. Phys. Chem. B*, **2007**, 111, 5602–5609; *Chem. Soc. Rev.*, **2010**, 39, 3210–3239). Due to the highest electronegativity, H_2PO_4^- dynamics would be accelerated by the electrostatic repulsion in CTF-TF to facilitate proton migration. Therefore, although the activation energy of H_3PO_4 @CTF-TF is high, the other positive factors still make it show excellent proton conductivity.

The corresponding explanation with references and Figures about how fluorine atoms enhance the proton conductivity is given in the revised manuscript as follows:

“Furthermore, the strength of the hydrogen bonding interactions can be probed by the values of binding energies from theoretical calculation. The H_3PO_4 molecules exhibit strong hydrogen bonding interaction when binding to the triazine nitrogen atoms of all three CTFs (Supplementary Fig. 14). Notably, getting benefit from the strong electron-withdrawing effect of fluorine, the nitrogen of triazine rings in CTF-TF exhibit stronger binding energy with H_3PO_4 as compared to that of CTF-1. In addition, as anchoring points, the fluorine atoms can not only provide more interaction sites, but also facilitate the proton dissociation between CTF-TF and H_3PO_4 due to $\text{F}\cdots\text{H}-\text{O}$ hydrogen bonds. The binding energy of triazine nitrogen to H_3PO_4 in CTF-TPA is the highest among them, but the amount of triazine nitrogen acceptors in the framework structures is halved as compared to that of CTF-1 and CTF-TF. Overall, the CTF-TF gives the highest strength of interaction with phosphoric acid among the three CTFs. The additional host-guest interaction sites given by F benefits for the phosphoric acid confinement and proton dissociation in the channels of CTF-TF, leading to better proton conductivity among the series.”

“The previous researches have shown that the phosphate anion (H_2PO_4^-) dynamics favors the long-range proton transport^{46,47}. As depicted in Supplementary Fig. 15, the electrostatic potentials (ESP) of the three CTFs exhibit distinct variations. The presence of fluorine (F) atoms in CTF-TF renders its pores electronegative, as indicated by a maximum ESP value of -0.533 eV. Conversely, the pores of CTF-1 exhibit electropositive characteristics, with an electronegative region observed near the nitrogen atoms of the triazine moieties. CTF-TPA

adopts a non-planar structure due to the incorporation of triphenylamine units, and an electronegative region is also observed at the N atoms of the triazine moieties. Consequently, CTF-TF exhibits a pronounced electron-withdrawing capacity within its pores, while CTF-1 displays the opposite behavior. CTF-TPA, on the other hand, exhibits electron-withdrawing or electron-donating capabilities at different positions within its pores, depending on the local environment. H_2PO_4^- dynamics that benefit for the proton conduction would be accelerated in CTF-TF due to electrostatic repulsion, while it may be inhibited in CTF-1 and CTF-TPA. Therefore, although the activation energy of H_3PO_4 @CTF-TF is high, the other positive factors still make it result in excellent proton conductivity.”

Supplementary Table 6. Summary of the binding energy and proton dissociation energy results of the theoretical calculation (PA: H_3PO_4).

	CTF-TF+PA (Triazine N...H)	CTF-TF+PA (F...H)	CTF-1+PA (Triazine N...H)	CTF-TPA+PA (Triphenyl- amine N...H)	CTF-TPA+PA (Triazine N...H)
Hirshfeld partition (IGMH)					Proton dissociation enthalpy (eV)	4.481	4.265	4.543	4.515	4.496
Binding energy (kcal/mol)	-18.767	-5.328	-16.150	-13.767	-22.347

Supplementary Fig. 14. Graphic representations of the binding energy of H_3PO_4 to various interaction sites for CTFs. (a) CTF-1(Triazine N), (b) CTF-TPA (Triazine N), (c) CTF-TPA (Triphenylamine N), (d) CTF-TF (F) and (e) CTF-TF (Triazine N). (H, white; P, brown; O, red;

N, blue; C, cyan; F, LT Magenta).

46. Traer, J. W., Britten, J. F. & Goward, G. R. A solid-state NMR study of hydrogen-bonding networks and ion dynamics in benzimidazole salts. *J. Phys. Chem. B* **111**, 5602-5609 (2007).

47. Asensio, J. A., Sánchez, E. M. & Gómez-Romero, P. Proton-conducting membranes based on benzimidazole polymers for high-temperature PEM fuel cells. A chemical quest. *Chem. Soc. Rev.* **39**, 3210–3239 (2010).

Supplementary Fig. 15. Electrostatic potential of (a) CTF-1, (b) CTF-TPA, and (c) CTF-TF.

Responses to Reviewer #2:

[Remarks to the Author]: Guan et al. present an interesting paper on the synthesis of fluorinated triazine framework that show good proton conductivity when impregnated with phosphoric acid. The work may generally be publishable but lacks the high degree of novelty and significance needed for Nature Communications. In addition, several issues are unclear (see below). Therefore, I do not recommend publication in this journal.

Response: Thank you so much for your great efforts in reviewing our revised manuscript. We sincerely appreciate your valuable comments and suggestions, which have certainly helped us to improve the quality of this manuscript. The manuscript was carefully revised based on these suggestions. Additional experiments and theoretical results were performed to provide solid supports in the revision. We look forward to your positive support for our revised manuscript.

To further strengthen the novelty and significance of our work, we explain them again in the following aspects:

(1) A New Method to Synthesize CTFs at Mild Conditions

The pursuit of clean energy solutions, such as water splitting and fuel cells, hinges on the development of highly stable proton-conducting materials operating at intermediate temperatures. Recently, CTF materials have emerged as a promising platform for such applications. Traditionally, fluorinated organic polymers, exemplified by NAFIONs, play a pivotal role in achieving high conductivity and stability. Consequently, it becomes paramount

to enhance the fluorine content within CTF materials. In this study, we introduce a novel approach to synthesize Covalent Triazine Frameworks (CTFs) through the direct cyclotrimerization of aromatic aldehydes, employing NH_4I as a readily available nitrogen source under mild conditions. Notably, this marks the first instance of synthesizing a type of perfluorinated CTF with a significantly high fluorine content through such gentle methods. Previous conventional approaches struggled to achieve this level of perfluorination under mild conditions. Thus, it is of great importance to develop new methods for CTFs with versatile structures, and this method provides a new avenue to create such functional CTFs. It is worth emphasizing that the starting materials and reagents used in this method are readily accessible, and the reaction conditions are notably mild, characterized by low temperatures (below $160\text{ }^\circ\text{C}$) and a relatively modest catalyst dosage.

(2) New Findings about the Effect of Fluorine in High Temperature Proton Conduction

Most of the works have studied the effect of some heteroatoms, such as nitrogen, in the proton conduction at high temperatures, but the effects of other special atoms have been rarely explored. Herein, we for the first time demonstrate that the fluorinated CTFs are promising for high temperature proton conduction. We show that the perfluorinated CTF endowed by the high fluorine content can provide as many anchoring sites as possible to interact with proton carriers and facilitate the proton transport. We further use DFT to investigate the host-guest interactions between H_3PO_4 and CTFs at a molecular level and revealed that the introducing of F atoms can not only directly increase the number of hydrogen bond acceptors for protons number, but also enhance the ability of proton dissociation and transport in the proton conduction process. Consequently, it is capable to exhibit an excellent proton conductivity ($1.82 \times 10^{-1}\text{ S cm}^{-1}$ at $150\text{ }^\circ\text{C}$) at high temperatures, superior to the most of porous materials reported to date (For example, $\text{H}_3\text{PO}_4@\text{COF-F6}$: $4.2 \times 10^{-2}\text{ S cm}^{-1}$, *J. Am. Chem. Soc.* **2020**, 142, 14357; $\text{H}_3\text{PO}_4@\text{TPB-DABI-COF}$: $1.52 \times 10^{-1}\text{ S cm}^{-1}$, *Angew. Chem. Int. Ed.* 2021, 60, 12918; $\text{H}_3\text{PO}_4@\text{NKCOF-54}$: $2.33 \times 10^{-2}\text{ S cm}^{-1}$, *Angew. Chem. Int. Ed.* 2022, e202217240). This work provides an alternative promising strategy for rational design of proton conducting media.

As above, the novel synthesis method for CTFs we have shown here, and the interesting effect of fluorine in the excellent proton conduction performance we revealed in this work, have not been reported in other works. Thus, we believe that this work shows sufficient novelty and advantage as compared to the reported references, which would be enlightening for the wide readership and thereby deserve to be published in Nature Communications.

To further clarify the novelty and advantage in this work, the manuscript has been thoughtfully revised, which is marked in the revision and summarized as followings.

✓ **Reorganization of Introduction section**

In the revision, the Introduction section is revised. The novelty and advantage of this new method are emphasized.

“The regulation of CTFs in proton conduction at high temperatures are mainly focused on heterocyclic nitrogen and aromatic structures. To enhance the proton conductivity, it is important to introduce as more hydrogen bond acceptors as possible, which may increase the host-guest interaction with the CTFs and proton carriers. However, due to the limitation of synthesis methods and the building blocks, the introduction of other precise hydrogen bond acceptors as active sites to increase the host-guest interaction in CTFs for proton conduction has not been widely explored.”

“While in CTFs, the introduction of fluorine atoms may greatly benefit for high temperature proton conduction because it may increase the host-guest interaction between the CTF and the proton carriers. In particular, perfluorinated CTFs with high fluorine content could exhibit the greatest number of anchoring sites with proton carriers in the framework structures.”

“Herein, we successfully established a new approach to synthesize CTFs via direct cyclotrimerization of aromatic aldehydes using NH₄I as facile nitrogen source for the first time (Fig.1). The conditions of this method are much milder, using low temperature (160 °C), open-air atmosphere, readily available raw materials, and relatively low catalyst dosage. We have found that the gentle approach is highly effective to synthesize CTFs with various structures. Particularly, the new synthetic approach enables the synthesis of perfluorinated CTF with high fluorine content and surface area under much milder conditions as compared to the reported methods (Supplementary Fig. 1 and Supplementary Table 1) ^{14,17-20}.

The resulting CTFs display promising proton conduction properties after binding with the H₃PO₄ with the merits of rich nitrogen content and high stability⁶⁻⁹. In particular, the electronegative fluorine sites together with the triazine units in the perfluorinated CTF (CTF-TF), which provide the precise host-guest interaction sites, can effectively lock H₃PO₄ and act as hydrogen bond acceptor to facilitate proton transport. Consequently, the CTF-TF loading with H₃PO₄ delivers an excellent proton conductivity of $1.82 \times 10^{-1} \text{ S cm}^{-1}$ at 150 °C, which ranks the highest value among the reported CTFs and is also comparable to other excellent porous organic polymer conductors. To the best of our knowledge, it is the first example to endow fluorinated CTF with high proton conduction by this work. Thus, this work provides a new powerful way to extend the functionality and applications of the CTFs.”

✓ **More results on mechanism of how fluorine atoms enhance the proton conductivity**

In the revision, DFT calculation was used to investigate the interaction between H₃PO₄

and CTFs. Due to the strong hydrogen bond between F and phosphoric acid, CTF-TF may act as hydrogen bond acceptor to stabilize the proton network and facilitate proton conduction. Thus, it gives the highest proton conductivity in the series.

“Furthermore, the strength of the hydrogen bonding interactions can be probed by the values of binding energies from theoretical calculation. The H_3PO_4 molecules exhibit strong hydrogen bonding interaction when binding to the triazine nitrogen atoms of all three CTFs (Supplementary Fig. 14). Notably, getting benefit from the strong electron-withdrawing effect of fluorine, the nitrogen of triazine rings in CTF-TF exhibit stronger binding energy with H_3PO_4 as compared to that of CTF-1. In addition, as anchoring points, the fluorine atoms can not only provide more interaction sites, but also facilitate the proton dissociation between CTF-TF and H_3PO_4 due to $\text{F}\cdots\text{H}-\text{O}$ hydrogen bonds. The binding energy of triazine nitrogen to H_3PO_4 in CTF-TPA is the highest among them, but the amount of triazine nitrogen acceptors in the framework structures is halved as compared to that of CTF-1 and CTF-TF. Overall, the CTF-TF gives the highest strength of interaction with phosphoric acid among the three CTFs. The additional host-guest interaction sites given by F benefits for the phosphoric acid confinement and proton dissociation in the channels of CTF-TF, leading to better proton conductivity among the series.”

Supplementary Fig. 14. Graphic representations of the binding energy of H_3PO_4 to various interaction sites for CTFs. (a) CTF-1(Triazine N), (b) CTF-TPA (Triazine N), (c) CTF-TPA (Triphenylamine N), (d) CTF-TF (F) and (e) CTF-TF (Triazine N). (H, white; P, brown; O, red; N, blue; C, cyan; F, LT Magenta). The $\text{N}(\text{triazine})\cdots\text{H}\cdots\text{O}$ hydrogen bonding between CTF-1 and H_3PO_4 yielded a binding energy of $-16.150 \text{ kcal mol}^{-1}$. The $\text{N}(\text{triazine})\cdots\text{H}\cdots\text{O}$ hydrogen bonding between CTF-TF and H_3PO_4 yielded a binding energy of $-18.767 \text{ kcal mol}^{-1}$, which is higher than that of the CTF-1, indicating the introducing of F can enhance the interaction. The binding energy of hydrogen bonding of CTF-TPA between N(triazine) and H_3PO_4 is $-22.346 \text{ kcal mol}^{-1}$. The binding energy for $\text{N}(\text{triphenylamine})\cdots\text{H}\cdots\text{O}$ in CTF-TPA is only $-13.767 \text{ kcal mol}^{-1}$, probably due to larger steric effect here that prevents the phosphoric acid molecule from

being close to the sites. Considering the amount of hydrogen bond acceptors (N) in the frameworks, the overall strength of the interaction between the CTF-TPA skeleton and H₃PO₄ should be smaller than CTF-1. The isosurfaces reveal the presence of extensive intermolecular interactions between the hydrogen donors (H₃PO₄) and acceptors (F atoms) in H₃PO₄@CTF-TF, indicating a strong interaction between H₃PO₄ and CTF-TF, when H₃PO₄ is inside the pore.

Supplementary Table 6. Summary of the binding energy and proton dissociation enthalpy results of the theoretical calculation (PA:H₃PO₄).

	CTF-TF+PA (Triazine N...H)	CTF-TF+PA (F...H)	CTF-1+PA (Triazine N...H)	CTF- TPA+PA (Triphenyl- amine N...H)	CTF- TPA+PA (Triazine N...H)
Hirshfeld partition (IGMH)					Proton dissociation enthalpy (eV)	4.481	4.265	4.543	4.515	4.496
Binding energy (kcal/mol)	-18.767	-5.328	-16.150	-13.767	-22.347

In the revision, the electronic structure and polarity of the CTFs was evaluated by electrostatic potential (ESP), which shows that the presence of F atoms in CTF-TF renders its pores electronegative, as indicated by a maximum ESP value of -0.533 eV. Proton transfer usually tends to occur between H₃PO₄ and H₂PO₄⁻ (*J. Electrochem. Soc.*, **2004**, 151, A8-A16; *Int. J. Quantum. Chem.*, **2011**, 111, 3212–3229). Phosphate anion (H₂PO₄⁻) dynamics contribute to long-range proton transport (*J. Phys. Chem. B*, **2007**, 111, 5602-5609; *Chem. Soc. Rev.*, **2010**, 39, 3210-3239). Due to electrostatic repulsion, H₂PO₄⁻ dynamics would be accelerated in CTF-TF, which facilitates the proton migration.

“The previous researches have shown that the phosphate anion (H₂PO₄⁻) dynamics favors the long-range proton transport^{46,47}. As depicted in Supplementary Fig. 15, the electrostatic potentials (ESP) of the three CTFs exhibit distinct variations. The presence of fluorine (F) atoms in CTF-TF renders its pores electronegative, as indicated by a maximum ESP value of -0.533 eV. Conversely, the pores of CTF-1 exhibit electropositive characteristics, with an

electronegative region observed near the nitrogen atoms of the triazine moieties. CTF-TPA adopts a non-planar structure due to the incorporation of triphenylamine units, and an electronegative region is also observed at the N atoms of the triazine moieties. Consequently, CTF-TF exhibits a pronounced electron-withdrawing capacity within its pores, while CTF-1 displays the opposite behavior. CTF-TPA, on the other hand, exhibits electron-withdrawing or electron-donating capabilities at different positions within its pores, depending on the local environment. H_2PO_4^- dynamics that benefit for the proton conduction would be accelerated in CTF-TF due to electrostatic repulsion, while it may be inhibited in CTF-1 and CTF-TPA. Therefore, although the activation energy of $\text{H}_3\text{PO}_4@\text{CTF-TF}$ is high, the other positive factors still make it result in excellent proton conductivity.”

46. Traer, J. W., Britten, J. F. & Goward, G. R. A solid-state NMR study of hydrogen-bonding networks and ion dynamics in benzimidazole salts. *J. Phys. Chem. B* **111**, 5602-5609 (2007).

47. Asensio, J. A., Sánchez, E. M. & Gómez-Romero, P. Proton-conducting membranes based on benzimidazole polymers for high-temperature PEM fuel cells. A chemical quest. *Chem. Soc. Rev.* **39**, 3210–3239 (2010).

Supplementary Fig. 15. Electrostatic potential of (a) CTF-1, (b) CTF-TPA, and (c) CTF-TF.

In the revision, to demonstrate the important role played by fluorine, we have adjusted the fluorine content by co-polymerizing. The BET surface area of CTF-TF-0.5 is measured to be $401.3 \text{ m}^2 \text{ g}^{-1}$, which is close to that of CTF-TF. It has a fluorine content of 18.92 wt%. Phosphoric acid content is controlled and same as CTF-TF (46%). Nevertheless, under the same conditions, the proton conductivity of $\text{H}_3\text{PO}_4@\text{CTF-TF-0.5}$ was tested to be $1.22 \times 10^{-1} \text{ S cm}^{-1}$ at $150 \text{ }^\circ\text{C}$, which demonstrated that fluorine is indeed responsible for enhancing the proton conductivity. The reason is that F atoms can act as hydrogen bond acceptor to stabilize the proton network and facilitate proton conduction.

“We also compared the CTF-TF with the CTF-TF-0.5 that is synthesized by adjusting the proportion of monomers. The surface area of CTF-TF-0.5 is close to that of CTF-TF ($401.3 \text{ m}^2 \text{ g}^{-1}$ vs. $407.7 \text{ m}^2 \text{ g}^{-1}$) (Supplementary Fig. 16), however, the proton conductivity of $\text{H}_3\text{PO}_4@\text{CTF-TF-0.5}$ ($1.22 \times 10^{-1} \text{ S cm}^{-1}$ at $150 \text{ }^\circ\text{C}$, Supplementary Fig. 17), is much lower than

that of $\text{H}_3\text{PO}_4@\text{CTF-TF}$, indicating that the fluorine here is a decisive factor for proton conductivity. The reason could be attributed to that the higher content of F atoms in the CTF pores may act as hydrogen-bonding acceptors⁴⁸, promoting the formation of hydrogen-bonding networks along the channel and facilitating proton transport⁴⁹”

48. Liu, Y. et al. Tough, stable and self-healing luminescent perovskite-polymer matrix applicable to all harsh aquatic environments. *Nat. Commun.* **13**, 1338 (2022).

49. Liu, S. et al. Construction of dense H-bond acceptors in the channels of covalent organic frameworks for proton conduction. *J. Mater. Chem. A* **11**, 13965-13970 (2023).

Supplementary Fig. 16. (a) synthesis of CTF-TF-0.5; (b) FT-IR spectra of CTF-TF-0.5; (c) N_2 adsorption and desorption isotherms (77 K) curves of CTF-TF-0.5.

Supplementary Fig. 17. Proton conductivity of CTF-TF-0.5 (a) Nyquist plots of $\text{H}_3\text{PO}_4@\text{CTF-TF-0.5}$; (b) Arrhenius plots for $\text{H}_3\text{PO}_4@\text{CTF-TF-0.5}$. The $\text{H}_3\text{PO}_4@\text{CTF-TF-0.5}$ exhibits proton conductivities of 4.37×10^{-2} , 5.10×10^{-2} , 5.97×10^{-2} , 6.78×10^{-2} , 8.08×10^{-2} , and $1.22 \times 10^{-1} \text{ S cm}^{-1}$ at 100, 110, 120, 130, 140 and 150 °C, respectively.

[1] The impedance measurements lack information about the relative humidity. Proton conductivity is known to be strongly dependent on humidity, including in organic polymers such as these. A careful control over and systematic variation of humidity would be mandatory to assess how much the conductivity is mediated by water.

Response: Thank you very much for the insightful comments. The control of humidity is indeed important in the low temperature (< 100 °C) proton conduction system, because the proton conductivity is mainly affected by the humidity at low temperature. Thus, the proton conduction under controlled humidity should be studied at lower temperatures (< 100 °C). Your suggestion is an important research topic and provides a direction for our next research.

However, in this study, what we investigated is the proton conductivity at higher temperatures (>100 °C, higher than the boiling point of water), which is different from the low temperature proton conduction systems. The high-temperature proton conduction employs high-boiling point proton carriers, i.e. $\text{H}_3\text{P}_3\text{O}_4$, which avoids the presence of water and thereby simplifies the water and heat management (*J. Power Sources*, **2013**, 231, 264–278). With an aim to develop efficient proton conductive materials at high temperatures, we report in this work a new synthetic route for CTF-TF under mild conditions and explore the effect of fluorine atoms in the high-temperature proton conductivity. Therefore, all the samples we have used have been efficiently vacuumed dried at high temperatures (150 °C) to efficiently remove the residual water before the measurement. And, the proton conductivity by EIS in our study was conducted under the anhydrous condition.

With all due respect, our experiments however have ultimately avoided the presence of water in the measurements, which makes it not accessible to control the humidity at high temperatures. According to the literatures, all the related reports have not investigated the controlled humidity in the high temperature proton conduction measurements (*Nat. Mater.* **2016**, 15, 722–726; *Adv. Energy Mater.* **2021**, 39, 2102300; *Nat. Commun.* **2020**, 11, 1981). Therefore, in this work the proton conductivity under controlled humidity at high temperatures (100~160 °C) is also not shown. We hope you could understand our situation.

For the clear understanding of the measurement condition, the related description of the measurement condition has been stated in the previous manuscript (proton conductivity measurement section in supplementary text):

“The proton conductivity (σ , S cm^{-1}) of CTFs- H_3PO_4 were characterized by EIS under the anhydrous condition.”

[2] What does "semi-crystalline" mean (XRD)? The term needs explanation. In addition, a peak in the XRD diagram (Suppl. fig. 5) at ca. 18° is not explained, either.

Response: Thank you for professional question. "Semi-crystalline" means that the materials comprise of both crystalline and amorphous regions (*PNAS* **2023**, 120, e2217363120). In order to avoid confusion for the readers, we have revised the description in revised manuscript.

"Powder X-ray diffraction (PXRD) measurements (Supplementary Fig. 5) indicated that the CTF-1 displays a low broad peak at 7.5°, it only contains low crystallinity to some degree."

And, the peak at about 18° in the PXRD pattern of CTF-1 should be attributed to the (210) crystal plane (*Adv. Mater.* **2019**, 31, 1807865; *Chem. Mater.* **2021**, 33, 1994–2003), which has been marked in revised **Supplementary Fig. 5**.

[3] The interpretation of the N₂ physisorption analysis is questionable. Hysteresis closure at ca. P/P₀ = 0.42 is most likely due to network effects ('forced hysteresis closure', 'tensile strength effect') rather than caused by mesoporosity.

Response: We appreciate this insightful comment. We completely agree with your professional comments. The hysteresis is usually attributed to the thermodynamic or network effects or the combination of these two effects (*Chem. Mater.* **2001**, 13, 3169–3183). H₄ hysteresis loops may merely arise from the presence of large mesopores embedded in a matrix with pores of much smaller size.

To address the Reviewer's concern, we have revised this part as follows:

"And it features a typical type H₄ hysteresis loop, which is most likely attributed to network effects, due to the presence of mesoporous and microporous structures together³⁵."

35. Kruk, M. Jaroniec, M. Gas adsorption characterization of ordered organic–inorganic nanocomposite materials. *Chem. Mater.* **13**, 3169-3183 (2001).

[4] According to the TGA data, the materials not really seem to be stable at temperature above ca. 150 °C.

Response: Thank you for your insightful comment. In order to clearly show the variation of CTFs at 100 ~ 200 °C, the TGA curve was locally enlarged (See graph below), and the results showed that CTFs could be stable at 150 °C without loss of mass. When the temperature further reach 200 °C, there is still only a negligible amount of mass loss in CTFs. The further weight loss at higher temperatures is also very normal, which could be due to the decomposition of some remaining or peripheral groups. Such phenomenon is widely observed for CTFs and other COFs as reported in the literatures (See figures below).

The partially enlarged TGA curves shows the CTF-TPA only has slight weight loss up to 200 °C.

Angew. Chem. Int. Ed.
2023, 62, e2023055

Angew. Chem. Int. Ed.
2023, 62, e2022161

J. Am. Chem. Soc.
2023, 145, 9520–9529

Nat. Commun. 2023, 14, 5097

Nat. Commun. 2023, 14, 3765

Nat. Mater. 2023, 22, 880–887

These are the TGA patterns of CTFs and other COFs reported by the previous literatures. As you may see that all of them showed some mass loss from 200 °C.

Responses to Reviewer #3:

[Remarks to the Author]: The manuscript by S. Jin and workers describes the low-temperature synthesis of covalent triazine frameworks and investigation of their proton conductivities. The synthesis approach is a good alternative to those reported procedures and the proton conducting performances of the obtained CTFs are high. I would like to recommend acceptance for publication after addressing the following points.

Response: We appreciate the reviewer for the insightful comments and positive evaluation.

[1] The authors should emphasize the novelty and advantage of the NH₄I-involved procedure and provide the comparison with other processes.

Response: We appreciate this comment. Following your suggestion, we emphasize the novelty and advantage of this work as follows:

Recently, CTF materials have emerged as a promising platform for high temperature proton conduction applications. To enhance the proton conductivity, it is highly desired to further increase the hydrogen bond acceptor sites and host-guest interaction with proton carriers. Also, to achieve excellent proton conductivity, it is important to retain intact of the chemical structure during synthesis and high content of the hydrogen bond acceptors in the framework, so it is strongly anticipated to synthesize the materials at lower temperatures. Traditionally, fluorinated organic polymers, exemplified by NAFIONs, play a pivotal role in achieving high conductivity and stability. Consequently, it becomes paramount to enhance the fluorine content within CTF materials.

In this study, we introduce a novel approach to synthesize Covalent Triazine Frameworks (CTFs) through the direct cyclotrimerization of aromatic aldehydes, employing NH_4I as a readily available nitrogen source under mild conditions. Notably, this marks the first instance of synthesizing a type of perfluorinated CTF with a significantly high fluorine content through such gentle methods. Previous conventional approaches struggled to achieve this level of perfluorination under mild conditions. Thus, it is of great importance to develop new method to construct CTFs with versatile structures under milder conditions.

We compare this method to other reported ones and highlight that the starting materials and the reagents in this method are all commercially available. The conditions of this method are quite mild, using low temperature ($< 160\text{ }^\circ\text{C}$), readily available raw materials, and relatively lower catalyst dosage. Based on this novel cyclotrimerization method, a highly fluorinated CTF (CTF-TF) can be successfully achieved under mild conditions as compared to previous methods. With this strategy, it is also potential to develop new and special functions for CTFs.

In the revision, the corresponding discussion to emphasize the novelty and advantage of this work in the Introduction section was revised as follows:

“The regulation of CTFs in proton conduction at high temperatures are mainly focused on heterocyclic nitrogen and aromatic structures. To enhance the proton conductivity, it is highly desired to introduce as more hydrogen bond acceptors as possible, which may increase the host-guest interaction with the CTFs and proton carriers. However, due to the limitation of synthesis methods and the building blocks, the introduction of other precise hydrogen bond acceptors as active sites to increase the host-guest interaction in CTFs for proton conduction has not been widely explored.”

“While in CTFs, the introduction of fluorine atoms may greatly benefit for high temperature proton conduction because it may increase the host-guest interaction between the CTF and the proton carriers. In particular, perfluorinated CTFs with high fluorine content could

exhibit the greatest number of anchoring sites with proton carriers in the framework structures.”

“Herein, we successfully established a new approach to synthesize CTFs via direct cyclotrimerization of aromatic aldehydes using NH_4I as facile nitrogen source for the first time (Fig.1). The conditions of this method are much milder, using low temperature ($160\text{ }^\circ\text{C}$), open-air atmosphere, readily available raw materials, and relatively low catalyst dosage. We have found that the gentle approach is highly effective to synthesize CTFs with various structures. Particularly, the new synthetic approach enables the synthesis of perfluorinated CTF with high fluorine content and surface area under much milder conditions as compared to the reported methods (Supplementary Fig. 1 and Supplementary Table 1) ^{14,17-20}”

[2] The proton conductivity results should be explained considering the chemical structures.

Response: We thank the constructive suggestion. By considering the chemical structures of CTFs, we have used DFT calculation to probe the interaction between H_3PO_4 and CTFs, to provide more insights into the role of fluorine atoms in modifying the electronic structure of the CTFs and discuss the mechanism of how fluorine atoms enhance the proton conductivity. Moreover, we constructed CTF-TF-0.5 with different contents of F atoms as control sample to prove that the high density of H-bond acceptors facilitates the formation of a proton transport pathway and thus improves the proton conductivity.

Firstly, to show the relationship of the chemical structures with the proton conductivity, the calculation of the binding energies between H_3PO_4 and CTFs at a molecular level using DFT was conducted. Due to the strong hydrogen bond between F and phosphoric acid, CTF-TF has the highest overall strength of interaction and the largest hydrogen bond acceptor density, which are beneficial to a high proton conductivity. According to the calculation results based on the different chemical structures, the CTF-TF exhibits higher binding energy (-18.676 kcal/mol) and lower proton dissociation energy (4.481 eV) on the N (triazine) sites as compared to CTF-1 (-16.150 kcal/mol and 4.543 eV), indicating the introducing of F atoms can change the polarity of the framework and indeed enhance the interaction between the CTFs and H_3PO_4 . Especially, the calculation shows that the F atoms have much lower proton dissociation energy (4.265 eV) as compared to the N atoms in the CTFs series, which further facilitates the proton dissociation and transport in the proton conduction. As for CTF-TPA, as compared to CTF-TF and CTF-1, it exhibits a higher binding energy of -22.347 kcal/mol with H_3PO_4 on the N (triazine) sites, but a lower binding energy on triphenylamine N sites. Despite its higher binding energy on the N (triazine) sites, the number of N (triazine) sites is only half of that in CTF-TF and CTF-1, which counteracts the positive effect of binding energy on the proton conduction.

Thus, the introducing of F atoms can not only directly increase the density of hydrogen bond acceptors for higher proton number, but also enhance the ability of proton dissociation and transport in the proton conduction process.

The related statement is added in the revised manuscript as follows:

“Furthermore, the strength of the hydrogen bonding interactions can be probed by the values of binding energies from theoretical calculation. The H_3PO_4 molecules exhibit strong hydrogen bonding interaction when binding to the triazine nitrogen atoms of all three CTFs (Supplementary Fig. 14 and Supplementary Table 6). Notably, getting benefit from the strong electron-withdrawing effect of fluorine, the nitrogen of triazine rings in CTF-TF exhibit stronger binding energy with H_3PO_4 as compared to that of CTF-1. In addition, as anchoring points, the fluorine atoms provide more interaction sites and facilitate the proton dissociation between CTF-TF and H_3PO_4 due to $\text{F}\cdots\text{H}-\text{O}$ hydrogen bonds. The binding energy of triazine nitrogen to H_3PO_4 in CTF-TPA is the highest among them, but the amount of triazine nitrogen acceptors in the framework structures is halved as compared to that of CTF-1 and CTF-TF. Overall, the CTF-TF gives the highest strength of interaction with phosphoric acid among the three CTFs. The host-guest interaction sites given by F benefits for the phosphoric acid confinement and proton dissociation in the channels of CTF-TF, leading to better proton conductivity among the series.”

Supplementary Fig. 14. Graphic representations of the binding energy of H_3PO_4 to various interaction sites for CTFs. (a) CTF-1(Triazine N), (b) CTF-TPA (Triazine N), (c) CTF-TPA (Triphenylamine N), (d) CTF-TF (F) and (e) CTF-TF (Triazine N). (H, white; P, brown; O, red; N, blue; C, cyan; F, LT Magenta). The N(triazine) $\cdots\text{H}\cdots\text{O}$ hydrogen bonding between CTF-1 and H_3PO_4 yielded a binding energy of $-16.150 \text{ kcal mol}^{-1}$. The N(triazine) $\cdots\text{H}\cdots\text{O}$ hydrogen bonding between CTF-TF and H_3PO_4 yielded a binding energy of $-18.767 \text{ kcal mol}^{-1}$, which is

higher than that of the CTF-1, indicating the introducing of F can enhance the interaction. The binding energy of hydrogen bonding of CTF-TPA between N(triazine) and H_3PO_4 is $-22.347 \text{ kcal mol}^{-1}$. The binding energy for N(triphenylamine)···H···O in CTF-TPA is only $-13.767 \text{ kcal mol}^{-1}$, probably due to larger steric effect here that prevents the phosphoric acid molecule from being close to the sites. Considering the amount of hydrogen bond acceptors (N) in the frameworks, the overall strength of the interaction between the CTF-TPA skeleton and H_3PO_4 should be smaller than CTF-1. The isosurfaces reveal the presence of extensive intermolecular interactions between the hydrogen donors (H_3PO_4) and acceptors (F atoms) in H_3PO_4 @CTF-TF, indicating a strong interaction between H_3PO_4 and CTF-TF, when H_3PO_4 is inside the pore.

Secondly, we calculated the electrostatic potentials (ESP) of the three CTFs, which shows that the chemical structures consist of different electronegativity in the pores. As shown in Supplementary Fig. 15, the ESP around the F atoms in CTF-TF has a value of -0.533 eV , indicating the F atoms renders its pores more electronegative. Proton transfer usually tends to occur between H_3PO_4 and H_2PO_4^- (*J. Electrochem. Soc.*, **2004**, 151, A8–A16; *Int. J. Quantum. Chem.*, **2011**, 111, 3212–3229). Phosphate anion (H_2PO_4^-) dynamics contribute to long-range proton transport (*J. Phys. Chem. B*, **2007**, 111, 5602 —5609; *Chem. Soc. Rev.*, **2010**, 39, 3210–3239). The different electronegativity arising from the CTFs chemical structures will cause the variable electrostatic repulsion between the CTFs and H_2PO_4^- . Therefore, the chemical structures of CTFs with different electronic structures, will cause the electronic repulsion with H_2PO_4^- , leading to the changing of its dynamics for proton migration. The most electronegative CTF-TF facilitates the dynamics of H_2PO_4^- , beneficial to the proton transfer, whereas electronegative CTF-TPA and CTF-1 are poorer.

The related statement was added in the revised manuscript as follows:

“The previous researches have shown that the phosphate anion (H_2PO_4^-) dynamics favors the long-range proton transport^{46,47}. As depicted in Supplementary Fig. 15, the electrostatic potentials (ESP) of the three CTFs exhibit distinct variations. The presence of fluorine (F) atoms in CTF-TF renders its pores electronegative, as indicated by a maximum ESP value of -0.533 eV . Conversely, the pores of CTF-1 exhibit electropositive characteristics, with an electronegative region observed near the nitrogen atoms of the triazine moieties. CTF-TPA adopts a non-planar structure due to the incorporation of triphenylamine units, and an electronegative region is also observed at the N atoms of the triazine moieties. Consequently, CTF-TF exhibits a pronounced electron-withdrawing capacity within its pores, while CTF-1 displays the opposite behavior. CTF-TPA, on the other hand, exhibits electron-withdrawing or electron-donating capabilities at different positions within its pores, depending on the local

environment. H_2PO_4^- dynamics that benefit for the proton conduction would be accelerated in CTF-TF due to electrostatic repulsion, while it may be inhibited in CTF-1 and CTF-TPA. Therefore, although the activation energy of $\text{H}_3\text{PO}_4@\text{CTF-TF}$ is high, the other positive factors still make it result in excellent proton conductivity.”

The related references and Figures were added in the revised manuscript as follows:

46. Traer, J. W., Britten, J. F. & Goward, G. R. A solid-state NMR study of hydrogen-bonding networks and ion dynamics in benzimidazole salts. *J. Phys. Chem. B* **111**, 5602-5609 (2007).

47. Asensio, J. A., Sánchez, E. M. & Gómez-Romero, P. Proton-conducting membranes based on benzimidazole polymers for high-temperature PEM fuel cells. A chemical quest. *Chem. Soc. Rev.* **39**, 3210–3239 (2010).

Supplementary Fig. 15. Electrostatic potential of (a) CTF-1, (b) CTF-TPA, and (c) CTF-TF.

Finally, to further probe the impact of the chemical structures, we constructed CTF-TF-0.5 with different contents of F atoms and proved that the high density of H-bond acceptors facilitates the formation of a proton transport pathway and thus improve the proton conductivity. CTF-TF-0.5 shows a BET surface area of $401.3 \text{ m}^2 \text{ g}^{-1}$, which is close to that of CTF-TF. It has a fluorine content of 18.92 wt%. Phosphoric acid content is controlled the same as CTF-TF (46%). Nevertheless, under the same conditions, the proton conductivity of $\text{H}_3\text{PO}_4@\text{CTF-TF-0.5}$ was tested to be $1.22 \times 10^{-1} \text{ S cm}^{-1}$ at $150 \text{ }^\circ\text{C}$, which demonstrates that fluorine is indeed responsible for enhancing the proton conductivity.

The related statements, references and supplementary figures are added in the revised manuscript as follows:

“We also compared the CTF-TF with the CTF-TF-0.5 that is synthesized by adjusting the proportion of monomers. The surface area of CTF-TF-0.5 is close to that of CTF-TF ($401.3 \text{ m}^2 \text{ g}^{-1}$ vs. $407.7 \text{ m}^2 \text{ g}^{-1}$) (Supplementary Fig. 16), however, the proton conductivity of $\text{H}_3\text{PO}_4@\text{CTF-TF-0.5}$ ($1.22 \times 10^{-1} \text{ S cm}^{-1}$ at $150 \text{ }^\circ\text{C}$, Supplementary Fig. 17), is much lower than that of $\text{H}_3\text{PO}_4@\text{CTF-TF}$, indicating that the fluorine here is a decisive factor for proton conductivity. The reason could be attributed to that the higher content of F atoms in the CTF

pores may act as hydrogen-bonding acceptors⁴⁸, promoting the formation of hydrogen-bonding networks along the channel and facilitating proton transport⁴⁹.”

48. Liu, Y. et al. Tough, stable and self-healing luminescent perovskite-polymer matrix applicable to all harsh aquatic environments. *Nat. Commun.* **13**, 1338 (2022).

49. Liu, S. et al. Construction of dense H-bond acceptors in the channels of covalent organic frameworks for proton conduction. *J. Mater. Chem. A* **11**, 13965-13970 (2023).

Supplementary Fig. 16. (a) synthesis of CTF-TF-0.5; (b) FT-IR spectra of CTF-TF-0.5; (c) N₂ adsorption and desorption isotherms (77 K) curves of CTF-TF-0.5.

Supplementary Fig. 17. Proton conductivity of CTF-TF-0.5 (a) Nyquist plots of H₃PO₄@CTF-TF-0.5; (b) Arrhenius plots for H₃PO₄@CTF-TF-0.5. The H₃PO₄@CTF-TF-0.5 exhibits proton conductivities of 4.37×10^{-2} , 5.10×10^{-2} , 5.97×10^{-2} , 6.78×10^{-2} , 8.08×10^{-2} , and 1.22×10^{-1} S cm⁻¹ at 100, 110, 120, 130, 140 and 150 °C, respectively.

[3] “The activation energy of H₃PO₄@CTF-1, H₃PO₄@CTF-TPA and H₃PO₄@CTF-TF are 0.22, 0.10 and 0.37 eV, respectively.” Please explain the reason behind the difference in activation energy.

Response: Thank you for the insightful suggestion. The reason behind the difference of

activation energies could be attributed to the different ratio contents of phosphate anion, which affects the energy barrier for proton conduction. The E_a is related to proton transfer pathways. Energy barriers for different proton transfer pathways heights in ascending order are (unit: kJ/mol): $\text{H}_3\text{PO}_4 \rightarrow \text{H}_2\text{PO}_4^-$ (~5.31); $\text{H}_4\text{PO}_4^+ \rightarrow \text{H}_3\text{PO}_4$ (~7.33); $\text{H}_3\text{PO}_4 \rightarrow \text{H}_4\text{P}_2\text{O}_7/\text{H}_3\text{PO}_4 \rightarrow \text{H}_3\text{PO}_4$ (~15.99) (*J. Electrochem. Soc.*, **2004**, 151, A8—A16; *Int. J. Quantum. Chem.*, **2011**, 111, 3212–3229). We revealed the species of H_3PO_4 in $\text{H}_3\text{PO}_4@\text{CTFs}$ using X-ray photoelectron spectroscopy (XPS), in which the phosphate anion (H_2PO_4^-) was significantly detected. Among these phosphate anions, the energy barriers for the proton transfer pathways that $\text{H}_3\text{PO}_4 \rightarrow \text{H}_2\text{PO}_4^-$ is almost the lowest in H_3PO_4 loaded polymers, so the $\text{H}_3\text{PO}_4 \rightarrow \text{H}_2\text{PO}_4^-$ mainly contributes to the proton transfer. According to the characterization of the XPS, it is found the proportion of phosphate anions (H_2PO_4^-) are in the order of $\text{H}_3\text{PO}_4@\text{CTF-TPA}$ (62.4%) > $\text{H}_3\text{PO}_4@\text{CTF-1}$ (61.0%) > $\text{H}_3\text{PO}_4@\text{CTF-TF}$ (56.3%). Therefore, the different activation energy could be attributed to the different content of the phosphate anion in the samples, which cause the different energy barriers to overcome. Thus, the higher the proportion of phosphate anion (H_2PO_4^-), the lower activation energy is required for proton transfer.

The corresponding statements, references and figures are added in the revised manuscript as follows:

“The E_a is related to proton transfer pathways. However, it is found that the activation energies are not directly correlated to the proton conductivities. We performed the XPS measurements for the three $\text{H}_3\text{PO}_4@\text{CTFs}$ and found that the high-resolution P 2p spectrums could be curve-fitted into two peaks at around 134.5 and 135.3 eV assigned to H_2PO_4^- and H_3PO_4 , respectively^{37,42} (Supplementary Fig. 13). The peak area ratios of H_2PO_4^- were calculated, which are in the order of $\text{H}_3\text{PO}_4@\text{CTF-TPA}$ (62.4%) > $\text{H}_3\text{PO}_4@\text{CTF-1}$ (61.0%) > $\text{H}_3\text{PO}_4@\text{CTF-TF}$ (56.3%). Because the energy barriers for proton transfer pathways via $\text{H}_3\text{PO}_4 \rightarrow \text{H}_2\text{PO}_4^-$ is the lowest in H_3PO_4 loaded proton conducting polymers^{43, 44}, the activation energy can be correlated to the H_2PO_4^- proportion. These results suggest that the higher the proportion of H_2PO_4^- , the lower the activation energy it may require for proton transfer. ”

37. Li, J., Wang, J., Wu, Z. Z., Tao, S. S. & Jiang, D. L. Ultrafast and stable proton conduction in polybenzimidazole covalent organic frameworks via confinement and activation. *Angew. Chem. Int. Ed.* **60**, 12918-12923 (2021).

42. Jiang, G. et al. Tuning the interlayer interactions of 2D covalent organic frameworks enables an ultrastable platform for anhydrous proton transport. *Angew. Chem. Int. Ed.* **61**, e202208086 (2022).

43. Ma, Y. L., Wainright, J. S., Litt, M. H. & Savinell, R. F. Conductivity of PBI

membranes for high-temperature polymer electrolyte fuel cells. *J. Electrochem. Soc.* **151**, A8-A16 (2004).

44. Li, S., Fried, J. R., Sauer, J., Colebrook, J. & Dudis, D. S. Computational chemistry and molecular simulations of phosphoric acid. *Int. J. Quantum. Chem.* **111**, 3212-3229 (2011).

Supplementary Fig. 13. High-resolution P 2p XPS spectra of (a) $\text{H}_3\text{PO}_4@CTF-1$, (b) $\text{H}_3\text{PO}_4@CTF-TPA$, and (c) $\text{H}_3\text{PO}_4@CTF-TF$.

[4] Section “Discussion” is not discussion, is the summary.

Response: Thanks so much for your insightful comment. According to the format of Nature Communications, this section should be “Discussion”. To make it different from the summary, we revised this section and have given more discussions in the revised manuscript as follows:

“This low-temperature method to construct CTFs by a cyclotrimerization reaction using aldehyde monomers and NH_4I as facile nitrogen source provides a new synthetic strategy for CTFs under mild conditions. By this novel approach, the perfluorinated CTF-TF with high fluorine content can be successfully achieved under much milder conditions as compared to the previous reports. Previously, many works have studied the effect of some heteroatoms, such as nitrogen, in the proton conduction at high temperatures, but the effects of other special atoms, such as fluorine atoms, have been rarely explored. In this work, we for the first time demonstrate that the fluorinated CTFs are promising for high temperature proton conduction. It is found that the perfluorinated CTF endowed by the high fluorine content can provide as many anchoring sites as possible to interact with proton carriers and facilitate the proton transport. By using the theoretical DFT calculation, we investigated the host-guest interactions between H_3PO_4 and

CTFs at a molecular level and revealed that the introducing of F atoms can not only directly increase the number of hydrogen bond acceptors for protons number, but also enhance the ability of proton dissociation and transport in the proton conduction process.”

[5] The optimum conditions for the synthesis of CTFs are not given. The yield provided in the text should be specified for some reaction temperature. All the yield results should be provided for every reaction temperature.

Response: Thank you for your comment. The optimum conditions are described in detail in the Methods section as follows. In the entire synthesis procedures, the temperature rise in the reactions are continuous processes. Therefore, only one total yield for the final product is given. In the optimization experiment, some yields at different temperature conditions are also supplemented to show the yield at every temperature condition (as shown in Supplementary Table 2, yellow marked).

The corresponding synthesis procedures with yields and the summary in Supplementary Table 2 are given in the revised manuscript as follows:

“Synthesis of CTF-1

CTF-1 was synthesized by a simple polycondensation route. 1,4-phthalaldehyde (134 mg, 1.0 mmol), NH_4I (290 mg, 2.0 mmol) and $\text{Fe}(\text{OAc})_3$ (85 mg, 0.45 mmol) were added to a solution of o-DCB (10 mL) in 25 mL round-bottom flask. The mixture was heated, with magnetic stirring, at 40 °C for 24 h, 80 °C for 24 h, 120 °C for 24 h and then 160 °C for 24 h. After cooling to room temperature, the solid was obtained by filtration, and then the filter cake was washed with DMF, dilute hydrochloric acid, water and absolute ethanol for several times, and then vacuum drying at 100 °C for 24 h to obtain a yellow powder sample. (116 mg, Yield: 87.8%).

Synthesis of CTF-TPA

4,4',4"-Nitrilotribenzaldehyde (165 mg, 0.50 mmol), NH_4I (290 mg, 2.0 mmol) and $\text{Fe}(\text{OAc})_3$ (56 mg, 0.3 mmol) were added to a solution of o-DCB (8 mL). The mixture was heated, with magnetic stirring, at 40 °C for 24 h, 80 °C for 24 h, 120 °C for 24 h and then 160 °C for 24 h. After cooling to room temperature, the solid was obtained by filtration, and then the filter cake was washed with DMF, dilute hydrochloric acid, water and absolute ethanol for several times, and then vacuum drying at 100 °C for 24 h to obtain a yellow-green powder sample. (122 mg, Yield: 74.8%).

Synthesis of CTF-TF

2,3,5,6-tetrafluoroterephthalaldehyde (103 mg, 0.50 mmol), NH_4I (150 mg, 1.0 mmol) and

Fe(OAc)₃ (45 mg, 0.25 mmol) were added to a solution of o-DCB(8 mL) in 25 mL round-bottom flask. The mixture was heated, with magnetic stirring, at 40 °C for 24 h, 80 °C for 24 h, 120 °C for 24 h and then 160 °C for 24 h. After cooling to room temperature, the solid was obtained by filtration, and then the filter cake was washed with DMF, dilute hydrochloric acid, water and absolute ethanol for several times, and then vacuum drying at 100 °C for 24 h to obtain a khaki powder sample. (85 mg, Yield: 82.5%).”

Supplementary Table 2. Synthesis of the CTF-1 under variable conditions.

Catalyst	Solvent	Temperature (°C)	Reaction time (h)	Yield [%]	S _{BET} [m ² /g]
FeCl ₃	Toluene	120	48	46	25
FeCl ₃	Mesitylene	120	48	33	31
FeCl ₃	o-DCB	120	48	61	42
FeCl ₃	DMSO	120	48	0	---
FeCl ₃	DMF	120	48	0	---
Fe(NO ₃) ₃	o-DCB	120	48	0	---
Fe(CF ₃ SO ₃) ₃	o-DCB	120	48	50	48
Fe ₂ (SO ₄) ₃	o-DCB	120	48	27	25
Fe(OAc) ₃	o-DCB	120	48	60	72
Cu(OAc) ₂	o-DCB	120	48	15	8
Fe(OAc) ₃	o-DCB	40	72	0	---
Fe(OAc) ₃	o-DCB	80	72	0	---
Fe(OAc) ₃	o-DCB	40°C, 24 h; 80°C, 24 h; 120°C, 24 h;		66	56
Fe(OAc) ₃	o-DCB	80°C, 24 h; 100°C, 24 h; 160°C, 24 h		76	100
Fe(OAc) ₃	o-DCB	40°C, 24 h; 80°C, 24 h; 120°C, 24 h; 160°C, 24 h		88	326.7

[6] The so-called host-guest interaction in CTFs, the authors should provide some more convinced evidence, experimental or theoretical results.

Response: Thank you for the insightful comment. Following your suggestions, we provide the evidences from both experimental and theoretical, which clearly show the host-guest interaction between the CTFs and phosphoric acid.

For experimental evidence, we studied the interaction between the CTF and the H_3PO_4 by XPS (Supplementary Fig. 4). After H_3PO_4 impregnation, the obvious peaks at around 400.9 eV in the deconvoluted N 1s spectra of $\text{H}_3\text{PO}_4@\text{CTF-1}$ (400.9 eV), $\text{H}_3\text{PO}_4@\text{CTF-TPA}$ (400.8 eV) and $\text{H}_3\text{PO}_4@\text{CTF-TF}$ (400.9 eV) could be observed, which indicates the formation of protonated pyridine nitrogen. The change in the binding energy indicates that the H_3PO_4 is locked into CTFs through strong interaction between the alkaline nitrogen and the H_3PO_4 . Specifically, there is a red-shift of F 1s peak (from 687.1 to 687.7 eV) observed in $\text{H}_3\text{PO}_4@\text{CTF-TF}$ (Supplementary Fig. 4d), indicating the electronegative fluorine binding sites in CTF-TF strongly interact with H_3PO_4 (Fig. 3d-f). We can also find that the high-resolution P 2p spectrums of the three $\text{H}_3\text{PO}_4@\text{CTFs}$ in the XPS measurements were curve-fitted into two peaks at around 134.5 and 135.3 eV, which can be assigned to H_2PO_4^- and H_3PO_4 , clearly showing the presence of the host-guest interactions.

The corresponding statements and discussion are added in revised manuscript as follows:

“To probe the reason for the excellent performance of CTF-TF in proton conductivities, we studied the interaction between the CTF and the H_3PO_4 by XPS (Supplementary Fig. 4). After H_3PO_4 impregnation, the obvious peaks at around 400.9 eV in the deconvoluted N 1s spectra of $\text{H}_3\text{PO}_4@\text{CTF-1}$ (400.9 eV), $\text{H}_3\text{PO}_4@\text{CTF-TPA}$ (400.8 eV) and $\text{H}_3\text{PO}_4@\text{CTF-TF}$ (400.9 eV) could be observed, which indicates the formation of protonated pyridine nitrogen^{37, 45}. The change in the binding energy indicates that the H_3PO_4 is locked into CTFs through strong interaction between the alkaline nitrogen and the H_3PO_4 . Specifically, there is a red-shift of F 1s peak (from 687.1 to 687.7 eV) observed in $\text{H}_3\text{PO}_4@\text{CTF-TF}$ (Supplementary Fig. 4d), indicating the electronegative fluorine binding sites in CTF-TF strongly interact with H_3PO_4 (Fig. 3d-f)^{4, 36}.”

To provide more evidence for the host-guest interaction, we further used theoretical calculations to probe the interaction between H_3PO_4 and CTFs. The strength of host-guest interaction can be estimated by comparing the values of binding energies. Notably, as anchoring sites, the fluorine atoms greatly enhance the interaction between CTF-TF and H_3PO_4 due to $\text{F}\cdots\text{H}-\text{O}$ hydrogen bonds. In addition, due to the strong electron-withdrawing effect of fluorine, the nitrogen of triazine rings in CTF-TF also exhibit stronger interaction with H_3PO_4 as

compared to that of CTF-1.

The corresponding statements and discussion are added in revised manuscript as follows:

“Furthermore, the strength of the hydrogen bonding interactions can be probed by the values of binding energies from theoretical calculation. The H_3PO_4 molecules exhibit strong hydrogen bonding interaction when binding to the triazine nitrogen atoms of all three CTFs (Supplementary Fig. 14 and Supplementary Table 6). Notably, getting benefit from the strong electron-withdrawing effect of fluorine, the nitrogen of triazine rings in CTF-TF exhibit stronger binding energy with H_3PO_4 as compared to that of CTF-1. In addition, as anchoring points, the fluorine atoms can not only provide more interaction sites, but also facilitate the proton dissociation between CTF-TF and H_3PO_4 due to $\text{F}\cdots\text{H}-\text{O}$ hydrogen bonds. The binding energy of triazine nitrogen to H_3PO_4 in CTF-TPA is the highest among them, but the amount of triazine nitrogen acceptors in the framework structures is halved as compared to that of CTF-1 and CTF-TF. Overall, the CTF-TF gives the highest strength of interaction with phosphoric acid among the three CTFs. The additional host-guest interaction sites given by F benefits for the phosphoric acid confinement and proton dissociation in the channels of CTF-TF, leading to better proton conductivity among the series.”

Supplementary Fig. 14. Graphic representations of the binding energy of H_3PO_4 to various interaction sites for CTFs. (a) CTF-1(Triazine N), (b) CTF-TPA (Triazine N), (c) CTF-TPA (Triphenylamine N), (d) CTF-TF (F) and (e) CTF-TF (Triazine N). (H, white; P, brown; O, red; N, blue; C, cyan; F, LT Magenta). The $\text{N}(\text{triazine})\cdots\text{H}\cdots\text{O}$ hydrogen bonding between CTF-1 and H_3PO_4 yielded a binding energy of $-16.150 \text{ kcal mol}^{-1}$. The $\text{N}(\text{triazine})\cdots\text{H}\cdots\text{O}$ hydrogen bonding between CTF-TF and H_3PO_4 yielded a binding energy of $-18.767 \text{ kcal mol}^{-1}$, which is higher than that of the CTF-1, indicating the introducing of F can enhance the interaction. The binding energy of hydrogen bonding of CTF-TPA between $\text{N}(\text{triazine})$ and H_3PO_4 is -22.346

kcal mol⁻¹. The binding energy for N(triphenylamine)···H···O in CTF-TPA is only -13.767 kcal mol⁻¹, probably due to larger steric effect here that prevents the phosphoric acid molecule from being close to the sites. Considering the amount of hydrogen bond acceptors (N) in the frameworks, the overall strength of the interaction between the CTF-TPA skeleton and H₃PO₄ should be smaller than CTF-1. The isosurfaces reveal the presence of extensive intermolecular interactions between the hydrogen donors (H₃PO₄) and acceptors (F atoms) in H₃PO₄@CTF-TF, indicating a strong interaction between H₃PO₄ and CTF-TF, when H₃PO₄ is inside the pore.

[7] Some typo and grammar errors should be corrected. The following are some examples.

the frameworks with more strengthened anchoring sites is (are) preferred

130 ppm ~145 ppm (130 ~145 ppm)

“Scanning electron microscope (SEM) and transmission electron microscope (TEM) were employed to observe the microscopic morphology”

Only the SEM results are employed to observe the morphology.

The proton conducting materials possessing good processability and flexibility is (are) desired.

The research of porous organic polymers in the field of proton conduction are (is) mostly focused on powder form interreact (interact) with H₃PO₄.

Response: Thank you very much for pointing out our omission. Following your comments, we have carefully corrected the mistakes in the revised manuscript. Moreover, the description of the morphology is rephrased as follows:

“Scanning electron microscope (SEM) were employed to observe the microscopic morphology of this series of polymers. Supplementary Fig. 8 show that CTFs are stacked nanosheets without special regular morphology. And transmission electron microscope (TEM) show CTF-TF displays more obvious thin layer structure (Supplementary Fig. 9).”

REVIEWER COMMENTS

Reviewer #1 (Remarks to the Author):

The authors have addressed most of the concerns of reviewer #1 and #2. However, the authors reported that H binds stronger with N than with F, which contradicts the general trend that F is more electronegative than N and therefore forms stronger hydrogen bonds. This finding needs to be explained or justified by the authors. I suggest that this manuscript can be accepted by Nature Communication after resolving this issue.

Reviewer #3 (Remarks to the Author):

The authors have addressed the comments and improved the manuscript except the host-guest interaction, which should be avoided to use in the manuscript. The authors should check the textbook of supramolecular chemistry. There is NOT host-guest interaction. In this case, it is not host-guest complexes. The interaction between CTFs' skeletons and phosphate is ion-dipole, or something like.

Response to Reviewer #1:

[Remarks to the Author]: The authors have addressed most of the concerns of reviewer #1 and #2. However, the authors reported that H binds stronger with N than with F, which contradicts the general trend that F is more electronegative than N and therefore forms stronger hydrogen bonds. This finding needs to be explained or justified by the authors. I suggest that this manuscript can be accepted by Nature Communication after resolving this issue.

Response: We appreciate for your recommendation of acceptance and the professional comment for this manuscript. The reason for the H binds stronger with N than with F is justified as follows.

In general, in the case of the proton donor group (D-H in D-H...A), the greater the electronegativity of D, the stronger the hydrogen bond. However, the correlation between electronegativity and the ability of hydrogen bond acceptors (A) is not clear, because electronegativity is a measure of the tendency to attract electrons, not protons (*Chem. Eur. J* **1997**, 3, 89-98). Usually, the fluoride ion may act as a strong proton acceptor, which can form the strongest hydrogen bonding in F-H...F, leading to the emphasis on electronegativity in the early accounts of H bonding (*J. Am. Chem. Soc.* **1964**, 86, 20, 4497). However, many researches have later found that despite the highest electronegativity of F, covalently bonded fluorine (C-F), as distinct from fluoride ion, are weaker hydrogen bond acceptors as compared to the C=N (*Tetrahedron*, **1966**, 52, 12613-12622; *Chem. Eur. J.* **2016**, 22, 7592–7601; *Acta Cryst.* **2017**, 73, 474–488; *Chem. Soc. Rev.*, **2008**, 37, 308-319; *Chem. Eur. J.* **1997**, 3, 89-98).

In this work, the C-F bond in the framework belongs to the class of organic fluorine. Due to the strongest electronegativity of F, the C-F bond is highly polarized. Benefited from the electrostatic attraction between the polarized C^{δ+} and F^{δ-} atom, the C–F bond is stable. However, this high polarized structure also suppresses the lone pair donation from fluorine atoms (*Chem. Soc. Rev.*, **2008**, 37, 308-319). Therefore, organic fluorine are usually weaker hydrogen bond acceptors than N.

The related statements and references are added in the revised Supplementary Information as follows:

“Although the covalently bonded fluorine are weaker hydrogen-bonding acceptors as compared to N atoms^{4,5}, the isosurfaces show that the introducing of fluorine atoms further increases the intermolecular interactions between the H₃PO₄ and triazine N sites. Moreover, the F atom sites can serve as additional anchoring sites for phosphoric acid, and contribute to the construction of hydrogen-bonding networks for efficient proton conduction.”

[4] C. Dalvit & A. Vulpetti. Weak intermolecular hydrogen bonds with fluorine: detection and implications for enzymatic/chemical reactions, chemical properties, and ligand/protein fluorine NMR screening. *Chem. Eur. J.* 22,7592–760 (2016).

[5] D. O'Hagan. Understanding organofluorine chemistry. An introduction to the C–F bond. *Chem. Soc. Rev.*, 37, 308-319 (2008).

Response to Reviewer #3:

[Remarks to the Author]: The authors have addressed the comments and improved the manuscript except the host-guest interaction, which should be avoided to use in the manuscript.

The authors should check the textbook of supramolecular chemistry. There is NOT host-guest interaction. In this case, it is not host-guest complexes. The interaction between CTFs' skeletons and phosphate is ion-dipole, or something like.

Response: We appreciate for your positive comment and professional suggestion. We agree with the Reviewer that the interactions involved in this work include hydrogen bonding interactions or ion-dipole interactions. Phosphoric acid molecules (H₃PO₄) interact with N and F in CTFs via hydrogen-bonding interactions, while the phosphate anion (H₂PO₄⁻) may interact with the CTF skeletons by ion-dipole interactions.

Following your suggestion, we removed or rephrased the corresponding places to avoid of using “host-guest interaction” in the revised manuscript. The corresponding places have been corrected in revised manuscript as follows:

“Covalent triazine frameworks (CTFs) are promising proton-conducting materials

at high temperatures but need more effective sites to strengthen **interaction** for proton carriers.”

“Due to the additional **hydrogen-bonding interaction** between fluorine atoms and proton carriers (H_3PO_4), the CTF-TF achieves an excellent proton conductivity of $1.82 \times 10^{-1} \text{ S cm}^{-1}$ at $150 \text{ }^\circ\text{C}$ after H_3PO_4 loading.”

“To enhance the proton conductivity, it is important to introduce as more hydrogen acceptors as possible, which may tune the **interaction** sites with the CTFs and proton carriers.”

“While in CTFs, the introduction of fluorine atoms may also greatly benefit for high temperature proton conduction because it may provide more **hydrogen-bonding interaction sites** between the CTF and the proton carriers.”

“In particular, the electronegative fluorine sites together with the triazine units in the perfluorinated CTF (CTF-TF), which provide the precise **interaction** sites, can effectively lock H_3PO_4 and act as hydrogen bond acceptor to facilitate proton transport.”

“With the variable building blocks which can tune the **interaction** with proton carriers in the present CTFs, we investigated the proton conductivity of the CTFs after being impregnated with phosphoric acid (85 %).”

“The additional **hydrogen-bonding interaction** sites given by F benefits for the phosphoric acid confinement and proton dissociation in the channels of CTF-TF, leading to better proton conductivity among the series.”

“By using the theoretical DFT calculation, we investigated the **interactions** between H_3PO_4 and CTFs at a molecular level and ...”

“The precision tuning of the **interaction between the framework and proton carriers** is vital to control proton conductivities for advancing proton conducting materials.”

The ion-dipole interaction is also mentioned in revised manuscript.

“ H_2PO_4^- dynamics that benefit for the proton conduction would be accelerated in CTF-TF due to electrostatic repulsion, while it may be inhibited in CTF-1 and CTF-TPA **owing to ion-dipole interactions.**”

REVIEWERS' COMMENTS

Reviewer #1 (Remarks to the Author):

I am satisfied with the authors' responses to my comments. I recommend that this manuscript be accepted for publication in Nature Communications.

Responses to Reviewer #1:

[Remarks to the Author]: I am satisfied with the authors' responses to my comments. I recommend that this manuscript be accepted for publication in Nature Communications.

Response: We appreciate for your recommendation of acceptance of this manuscript.